# Revisiting LRP: Positional Attribution as the Missing Ingredient for Transformer Explainability

**Yarden Bakish**
Tel-Aviv University

**Itamar Zimerman**
Tel-Aviv University

**Hila Chefer**
Tel-Aviv University

**Lior Wolf**
Tel-Aviv University

## Abstract

The development of effective explainability tools for Transformers is a crucial pursuit in deep learning research. One of the most promising approaches in this domain is Layer-wise Relevance Propagation (LRP), which propagates relevance scores backward through the network to the input space by redistributing activation values based on predefined rules. However, existing LRP-based methods for Transformer explainability entirely overlook a critical component of the Transformer architecture: its positional encoding (PE), resulting in violation of the conservation property, and the loss of an important and unique type of relevance, which is also associated with structural and positional features. To address this limitation, we reformulate the input space for Transformer explainability as a set of position-token pairs. This allows us to propose specialized theoretically-grounded LRP rules designed to propagate attributions across various positional encoding methods, including Rotary, Learnable, and Absolute PE. Extensive experiments with both fine-tuned classifiers and zero-shot foundation models, such as LLaMA 3, demonstrate that our method significantly outperforms the state-of-the-art in both vision and NLP explainability tasks. Our code is publicly available.

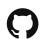 https://github.com/YardenBakish/PE-AWARE-LRP

## 1   Introduction

Explainable AI (XAI) is increasingly vital in deep learning (DL), where models often achieve remarkable performance but operate as opaque "black boxes" [7, 16]. This lack of transparency reduces trust, limits user engagement, and complicates troubleshooting, thereby restricting the use of DL in applications where decision-making transparency is essential. Consequently, developing XAI techniques for DL models has become an important research domain [29]. This task, however, is challenging, due to the inherent complexity of these models, which cannot be easily represented by simple functions.

Transformer-based architectures, which have become dominant in DL, present additional challenges for explainability due to their large scale, often containing billions of parameters. To address this, researchers have developed various attribution methods specifically designed for Transformers [14, 2, 3, 1]. Among these, model-specific XAI techniques have gained prominence, providing explanations based on the model's parameters, internal representations, and overall architecture.

The most effective model-specific XAI techniques, and the current state-of-the-art for Transformer explainability, are LRP-based, such as [2]. LRP is a well-established attribution technique that explains a model's predictions by propagating relevance scores backward through the network, redistributing activation values based on predefined propagation rules. Unlike gradient-based methods, which often suffer from issues like vanishing gradients or numerical instabilities, LRP provides a more stable and precise way to trace how information flows through each layer.

39th Conference on Neural Information Processing Systems (NeurIPS 2025).

Recently, several refinements have been proposed to improve the stability and faithfulness of LRP rules for Transformers, leading to more robust and reliable interpretability techniques. Notable examples include [3] and [2], which introduce custom rules for propagating LRP through attention mechanisms, layer normalization, and other key components. Despite these advancements, we identify a critical gap in this extensive line of work: all existing LRP-based methods for Transformers overlook the need for PE-aware LRP rules and do not propagate attribution through positional encoding. This omission results in the loss of a key aspect of relevancy related to positional concepts, limiting the ability to provide faithful and comprehensive explanations.

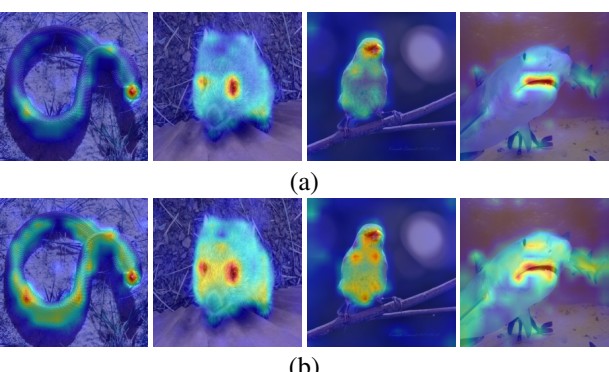

Figure 1: **(a)** Explainability heatmaps of Attention-LRP (AttnLRP) [2], which is the state-of-the-art LRP for transformer explainability. **(b)** The LRP heatmap obtained from the positional part that is ignored by the existing LRP methods, including AttnLRP. The relevancy captured by the PE component of the transformer is less sparse and captures more of the object.

To mitigate this problem, we propose Positional-Aware LRP (PA-LRP), a novel technique that significantly improves upon previous methods through two fundamental modifications: (i) Reformulating the input space of the Transformer explainability problem to incorporate positional information. Instead of relying solely on the vocabulary space, we define the input space as a set of position-token pairs. (ii) Introducing the first LRP rules specifically designed to propagate relevance across standard positional encoding (PE) layers, including learned PE, Rotary PE [33], and others. To enhance stability and faithfulness, our rules are further improved through techniques such as reparameterization of PE layers, linearization, and defining an appropriate sink for positional relevance to ensure that position-associated information is properly absorbed, which we validate to be crucial for precise propagation. Moreover, we provide a complementary theoretical analysis to prove that our rules do not violate the conservation property.

**Our main contributions** consist of the following: (i) We identify a critical gap in current LRP-based XAI techniques for Transformers: they overlook the attribution of positional encodings (PE). This omission results in a violation of the conservation property for input-level PE, as shown in Lemma 3.1, and leads to unfaithful heatmaps when handling positional features, as demonstrated in Lemma 3.3. We empirically validate that this omission is a critical limitation by significantly outperforming existing methods, and demonstrating that in certain cases, assigning relevance to PE alone can surpass standard SoTA Transformer explainability techniques, showing that this signal is significant, as shown in Tables 3–5 and Figure 3. A Additionally, the obtained signal is complementary and distinct from the non-positional signal, better capturing spatial, positional, and structural relationships, as shown in Figure 1 (ii) We introduce PA-LRP, a theoretically grounded and PE-aware technique for assigning relevance in Transformers. PA-LRP significantly outperforms previous methods across both fine-tuned classifiers and zero-shot foundation models, in both NLP and vision tasks, on multiple models such as LLaMA 3, DeiT, and others. (iii) Providing an open-source and user-friendly implementation of our method, along with demos and practical examples, to facilitate adoption by the broader research and practitioner community.

## 2 Background and Related Work

In this section, we describe the scientific context for discussing LRP-based Transformer explainability, along with the necessary terminology and symbols needed to describe our method.

### 2.1 Positional Encoding in Transformers

Transformer-based [37] architectures rely on self-attention, which computes contextual relationships between tokens using:

$$\text{Attention}(X) = \text{Softmax}\left(\frac{QK^T}{\sqrt{d_k}}\right)V \tag{1}$$

where, $K = XW_K, Q = XW_Q, V = XW_V$ represent key, query, and value matrices respectively, $d_k$ is the embedding dimension, and $W_Q, W_K, W_V$ are learnable linear projection matrices.

This attention mechanism is duplicated over several "heads" and is wrapped by standard DL peripherals such as Layer Normalization, FFNs, and skip connections, forming the core structure of a Transformer model by:

$$X' = \text{LayerNorm}\,(X + \text{Attention}(X))\,, \quad X'' = \text{LayerNorm}\,(X' + \text{FFN}(X')) \qquad (2)$$

where *FFN* applies a two-layer linear transformation with activations in the middle.

Transformers operate on sets of tokens rather than ordered sequences, making them permutation-invariant by design. Unlike architectures with built-in order sensitivity such as RNNs [21, 23], Transformers require explicit positional encoding (PE) to capture sequence structure. PE can be introduced at different stages of the model: it can be added to token embeddings at the input layer, as seen in learnable PE and sinusoidal PE [37], or integrated within the attention mechanism at each layer, as employed in Rotary PE (RoPE) [33] and Alibi [26]. The key insight of this paper is that while PE is well known for its important role in the forward pass [17], its crucial role in propagation-based XAI methods, such as LRP, has been largely overlooked, leading to violations of conservation and the loss of significant relevance, which often carries distinctive positional and structural meanings.

**Learnable PE.**    Learnable PE represents positions as trainable parameters, allowing the model to learn position representations directly from data. This approach offers flexibility and adaptability.

**Sinusoidal PE.**    Sinusoidal PE, introduced in the original Transformer model [37], encodes positions using sine and cosine functions with different non-trainable frequencies. Because it is based on absolute positions, it is less effective in tasks where relative positional information is more important.

**Rotary PE (RoPE).**    RoPE [33] incorporates positional information by rotating token embeddings in a structured manner, enabling the model to naturally encode relative positions. Specifically, each key and query vector is transformed using a per-position block-diagonal rotation matrix. Unlike learnable or sinusoidal PEs, RoPE encodes relative positional relationships through the multiplication of rotation matrices. Due to its effectiveness, many popular LLMs, including SAM2 [28],Pythia [12], LLaMA [36], Qwen [10], Gemma [35], and others are built on top or RoPE.

Other PE techniques, such as ALiBi [26] and relative PEs [31, 27], are described in Appendix A.

## 2.2   Model-Specific XAI and LRP

Methods for explaining neural models have been extensively studied in the context of DNNs, particularly in NLP [6, 39] and computer vision [30]. A widely adopted strategy for this task is the use of model-specific techniques, which exploit the internal architecture and parameters of neural models to generate explanations. One notable method in this category is LRP [8], which propagates relevance scores, denoted by $\mathcal{R}(\cdot)$, backwards through the network by redistributing activation values. Propagation relies on predefined rules and interactions between tokens.

**LRP.**    LRP is an evolution of gradient-based methods, such as Input $\times$ Gradient [32, 9], which often suffer from issues like numerical instabilities and gradient shattering [11]. LRP enhances backpropagation rules by enforcing two key principles: (i) the conservation property, which ensures that the total relevance is preserved across layers. Namely, for a layer $M$, where $Y = M(X)$, the relevance of the output $\mathcal{R}(Y)$ is equal to the relevance of the input $\mathcal{R}(X)$. (ii) The prevention of numerical instabilities during propagation. To achieve these goals, LRP rules are often derived from the Deep Taylor Decomposition principle [25], redistributing relevance scores at each layer based on the first-order Taylor expansion of the layer's function.

## 2.3   XAI for Transformers

The first model-specific XAI methods for Transformers were based on attention maps [13, 15], leveraging attention scores to quantify the contribution of each token to others across layers. Building on this approach, Abnar and Zuidema [1] introduced the attention rollout technique, which aggregates attention matrices across multiple layers to provide a more holistic explanation. However, Jain and

Wallace [22] later demonstrated that attention-based techniques can be misleading, as attention scores do not always correlate with gradient-based feature importance measures or actual model behavior. To address these limitations, Chefer et al. [14]. developed a hybrid XAI method that combines LRP scores with attention maps, marking a breakthrough in the field by improving attribution fidelity.

Purely LRP-based XAI methods for Transformers were first introduced in [38] and later refined by Ali et al. [3], who developed custom LRP rules tailored for LayerNorm and attention layers to preserve conservation properties and ensure numerical stability. More recently, Achtibat et al. [2] further improved this approach by designing more faithful propagation rules for self-attention, achieving state-of-the-art performance in Transformer explainability. To the best of our knowledge, this represents the most advanced technique in the field and serves as our primary baseline.

Interestingly, despite extensive research in this area, none of these approaches propagate relevance through PE layers. This omission leads to a loss of significant relevance associated with positional and structural features, ultimately resulting in less faithful and holistic attributions.

## 3 Method

In this section, we describe our PE-aware LRP rules. We first revise the input space used in the Transformer explainability problem in Section 3.1. Then, building upon this formulation, we define our custom LRP rules in Section 3.2 and 3.3. Finally, in Section 3.4 we provide theoretical analysis of our method, showing that our PE-aware LRP rules are theoretically grounded.

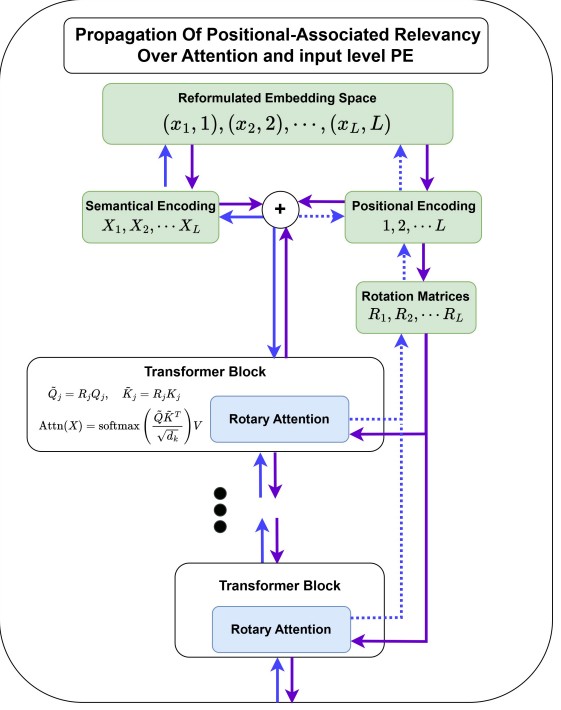

Figure 2: Visualization of our method for propagating PE-associated relevance. Blue arrows indicate the forward path, while purple arrows represent the LRP propagation rules. Dashed arrows denote custom position-aware rules defined in our method.

### 3.1 Reformulating the Vocabulary Space

To comprehensively attribute positional information, we must define a **sink** that absorbs position-associated relevance. To achieve this, we reformulate the explainability problem for Transformers. Given an embedding matrix $E \in \mathbb{R}^{|V| \times D}$, where $|V|$ is the number of tokens in the vocabulary $V$, and $D$ is the embedding size, previous methods traditionally define the input space $\mathcal{S}$ as

$$\mathcal{S} = \{E_i \mid E_i \in \mathbb{R}^D\}, \qquad (3)$$

where $E_i$ is the $i$-th row of the embedding matrix $E$.

In contrast, we reformulate the input space in the following manner:

$$\mathcal{S} = \{(E_i, P_{j,1}, P_{j,2}, \cdots, P_{j,K}) \mid E_i \in \mathbb{R}^D, j \in [L], k \in [K], P_{j,i} \in \mathbb{R}^{D'}\} \qquad (4)$$

where $L$ is the sequence length, $D'$ is the dimension of the positional embeddings, and K is the number of layers. Thus, in our formulation, each token in the input space consists of two ingredients: one representing the per-layer positional embedding $P_{j,k}$ for all layers $k \in [K]$, and the other representing the semantic embedding $E_i$.

We define a separate sink for positional relevance at each attention layer to ensure that the omission of certain positional features in one layer does not obscure or override essential features and relevance captured in other layers, and to validate that important positional attributions are not discarded.

Building upon the formulation of Eq. 4, the next two sections define LRP rules that enable stable propagation of relevance from standard positional encoding techniques to $j \in [L]$: Section 3.2 discusses input-level PE, while Section 3.3 covers attention-level PE.

## 3.2 LRP-rules for Input Level PE

We begin with the simplest form of positional encoding—learnable PE—and then demonstrate that other input-level PEs can be reparameterized in a similar manner. For brevity, we assume that $P_j$ is a vector rather than a matrix, namely $P_j = P_{j,1}$. We also tie the embedding dimensions of both the semantical and positional vectors ($D = D'$).

**Learnable PE.** This layer learns positional information during training through an embedding matrix $P \in \mathbb{R}^{L \times D}$ where $D$ represents the embedding dimension of positional information, and $L$ denotes the maximum sequence length. For each sample, the positional and semantic embeddings are summed to obtain the final input representation. Formally, the combined embedding for the token at position $j$ with token index $i$ is given by $P_J + E_i$. Thus, we can propagate relevance from the input of the first transformer block $\mathcal{R}(z_i)$, to the positional component $P_j$ of token $j$ by using the standard LRP epsilon rule for addition [2]:

$$\text{PA-LRP for input-level PE}: \mathcal{R}(P_j) = P_j \frac{\mathcal{R}(z_i)}{P_j + E_i + \epsilon} \tag{5}$$

**Sinusoidal PE.** This method encodes position information via a unique vector of sine and cosine values constructed by:

$$\text{Sinusoidal PE(j)}[2i] = \sin\left(\frac{j}{10000^{\frac{2i}{D}}}\right), \quad \text{Sinusoidal PE(j)}[2i+1] = \cos\left(\frac{j}{10000^{\frac{2i}{D}}}\right) \tag{6}$$

Thus, the values derived from Eq. 6 can be used to reparameterize the embedding matrix $E$, replacing the learned vectors with their corresponding sine and cosine values. Such reparameterization eliminates the need to propagate gradients through non-linear functions such as sine and cosine, improving efficiency and stability.

## 3.3 LRP-rules for Attention-level PE

For attention-level PE, we focus on describing the PA-LRP rules for RoPE [33] as a representative example. For the derivation of the PA-LRP rules for ALiBi [26], we refer the reader to Appendix B. At each layer $k$, RoPE modifies the queries (Q) and keys (K) matrices before computing the attention scores. This modification is done by multiplying each key and query vector by a position-dependent rotation matrix $R_{j,k} \in \mathbb{R}^{D \times D}$ where $j \in [L]$. The rotation matrix is a block-diagonal matrix defined as follows:

$$\forall j \in [L], k \in [K] : R_{j,k} = \begin{bmatrix} \cos\theta_j^{(1)} & -\sin\theta_j^{(1)} & \ldots & 0 & 0 \\ \sin\theta_j^{(1)} & \cos\theta_j^{(1)} & \ldots & 0 & 0 \\ \vdots & \vdots & \ddots & \vdots & \vdots \\ 0 & 0 & \ldots & \cos\theta_j^{(D/2)} & -\sin\theta_j^{(D/2)} \\ 0 & 0 & \ldots & \sin\theta_j^{(D/2)} & \cos\theta_j^{(D/2)} \end{bmatrix} \tag{7}$$

where each rotation angle $\theta_j^{(m)}$ is defined as $\theta_j^{(m)} = j\omega_m$, where $\omega_i = 10000^{\frac{D}{2(m-1)}}$.

Note that in RoPE, as in other attention-level positional encodings, the positional information is represented by a matrix $R_{j,k}$. Accordingly, we assume: $P_{j,k} = \text{Flattening}(R_{j,k})$, $D' = D^2$. Thus, we can propagate relevance from $\mathcal{R}(R_{j,k})$ to $\mathcal{R}(P_{j,k})$ by unflattening the relevance.

Now, a key remaining step is to define how relevance should be propagated to $R_j$. The RoPE computation is executed before computing the attention scores, transforming the per-position queries and keys are as follows:

$$\forall j \in [L] : \tilde{\boldsymbol{Q}}_j = R_j \boldsymbol{Q}_j, \quad \tilde{\boldsymbol{K}}_j = R_j \boldsymbol{K}_j, \quad \text{Rotary Attention}(X) = \text{Softmax}\left(\frac{\tilde{\boldsymbol{Q}}\tilde{\boldsymbol{K}}^T}{\sqrt{d_k}}\right)V . \tag{8}$$

Our formulation builds on top of AttnLRP [2], which propagates relevance over the queries $\tilde{\boldsymbol{Q}}$ and keys $\tilde{\boldsymbol{K}}$, resulting in their corresponding relevance scores $\mathcal{R}(\tilde{\boldsymbol{Q}}), \mathcal{R}(\tilde{\boldsymbol{K}})$. To propagate relevance

from these matrices to the rotation matrices $R_j$, we apply the standard uniform-LRP rule for matrix multiplication separately to each key and query, then summing both terms to produce a final attribution map per- attention layer, as follows:

$$\forall j \in [L] : \mathcal{R}(R_j) = \frac{1}{2}\mathcal{R}(\tilde{\boldsymbol{Q_j}}) + \frac{1}{2}\mathcal{R}(\tilde{\boldsymbol{K_j}}) \tag{9}$$

Up to this point, we have described the PA-LRP rules for a single attention layer. However, transformer-based models stack $M$ transformer blocks. Thus, we interpret the positional information as a vector that is passed from the input to all attention blocks via a semi-skip connection mechanism, as illustrated in Figure 2. This interpretation explains why without propagating relevance across PE layers, some of the relevance is lost, leading to unfaithful explanations that ignore position-associated aspects. Consequently, positional relevance from all layers is aggregated according to the LRP addition rule, similar to skip connections.

**Overall Method.** Our PA-LRP rules allow us to assign relevance to the positional part of the input space. For the non-positional part, we use the same rules as defined in AttnLRP [2]. Finally, we aggregate the relevance scores by summing their corresponding absolute values across feature dimensions, similar to previously proposed methods. It is worth noting that although our rules are built on top of the AttnLRP framework, they are not limited to it. Our input-level PE rules can be decoupled and applied to any LRP method, while the attention-level PE rules can be integrated with alternative formulations, as long as they propagate relevance through the attention matrices and preserve the connection between PE and the computational graph.

As a result, similar to other LRP methods, our approach can produce explainability maps with computational efficiency comparable to a single backward pass. We further clarify that although our method introduces several modifications in the forward path and input space, it does not require any changes to the transformer itself. Instead, these modifications propose an equivalent forward path that allows us to better define the propagation rules.

### 3.4 Theoretical Analysis

To support our PA-LRP rules, we now provide theoretical evidence demonstrating that they satisfy the key LRP criteria. First, the following two lemmas prove that our proposed LRP rules satisfy the conservation property.

**Lemma 3.1.** *For input-level PE transformers, the conservation property is violated when disregarding the positional embeddings' relevancy scores.*

**Lemma 3.2.** *For attention-level PE transformers, our PE-LRP rules satisfy the conservation property.*

Next, we present a lemma based on a key example illustrating that existing methods exhibit low faithfulness. In particular, we show that within simplified settings, LRP yields unfaithful explanations when the task relies heavily on positional features, such as predicting the number of tokens.

**Lemma 3.3.** *For attention-level PE transformers, current LRP attribution rules achieve low faithfulness, especially when considering positional features.*

The proofs and examples are detailed in Appendix E.

## 4   Experiments

To assess the effectiveness of our PA-LRP rules, we perform a comprehensive set of experiments across both Vision and NLP domains. First, in Section 4.1, we conduct experiments in the Vision domain using DeiT, including perturbation and segmentation tests. Next, in Section 4.2, we perform perturbation tests, ablation study, and conservation analysis in NLP. Test results are reported in each subsection, whereas the complete statistical analysis, including variance measures and paired t-test scores, is provided in Appendix I.

We begin by describing our baselines, ablation variant, and evaluation metrics:

**Baseline and Ablation Variant.** Our primary baseline for comparison is AttnLRP [2], as it represents the SoTA in general transformer XAI, and our method builds on top of it for non-positional

components. The key distinction between our approach and this baseline (as well as other LRP-based methods) is our ability to attribute relevance to positional information. Our composite approach that balances both positional and non-positional relevance is denoted as PA-LRP, or 'ours'. Additionally, to isolate the effect of the positional encoding, we introduce an ablation variant denoted by 'PE Only', which directly measures the relevance assigned to positional components at the input space using our custom attribution rules.

Although empirical evaluation of attribution methods is inherently challenging, we validate our PA-LRP method using perturbation and segmentation tests. Below, we describe these metrics:

**Perturbation Tests.** Perturbation tests are split into two metrics: positive and negative perturbations, which differ in the order in which pixels or tokens are masked. In positive perturbation, pixels or tokens are masked in descending order of relevance. An effective explanation method identifies the most influential regions, leading to a noticeable drop in the model's score (measured in comparison to the predicted or target class) as these critical areas are gradually removed. In negative perturbation, masking begins with the least relevant elements and progresses toward more important ones. A reliable explanation should keep the model's prediction stable, demonstrating robustness even when unimportant components are masked.

Following [3, 41], in both Vision and image domains, the final metric is quantified using the Area-Under-Curve (AUC), capturing model accuracy relative to the percentage of masked pixels or tokens, from 10% to 90%. For further technical details, see Appendix H.

**Segmentation Tests.** For attribution methods in vision, segmentation tests are a set of evaluations used to assess the quality of a model's ability to distinguish foreground from background in an image.

These tests compare the labeled segmentation image, which indicates whether each pixel belongs to the background or the foreground, with the explainability map after it has been binarized using a thresholding technique. Then, several metrics are computed over both images: (i) Pixel Accuracy: The percentage of correctly classified pixels, measuring how well the predicted segmentation aligns with the ground truth. (ii) Mean Intersection-over-Union (mIoU): The ratio of the intersection to the union of the predicted and ground-truth segmentation maps, averaged across all images. (iii) Mean Average Precision (mAP): A metric that considers precision and recall trade-offs at different thresholds, providing a robust assessment of segmentation quality.

### 4.1 Results for Vision Transformers

For vision models, we present both quantitative and qualitative analysis.

**Qualitative Analysis.** For qualitative analysis, we visualize the explainability maps obtained from our method, the AttnLRP [2] baseline, and the ablation variant that focuses exclusively on PE-associated relevance denoted by PE Only. Additional examples with larger images are presented in appendix C.

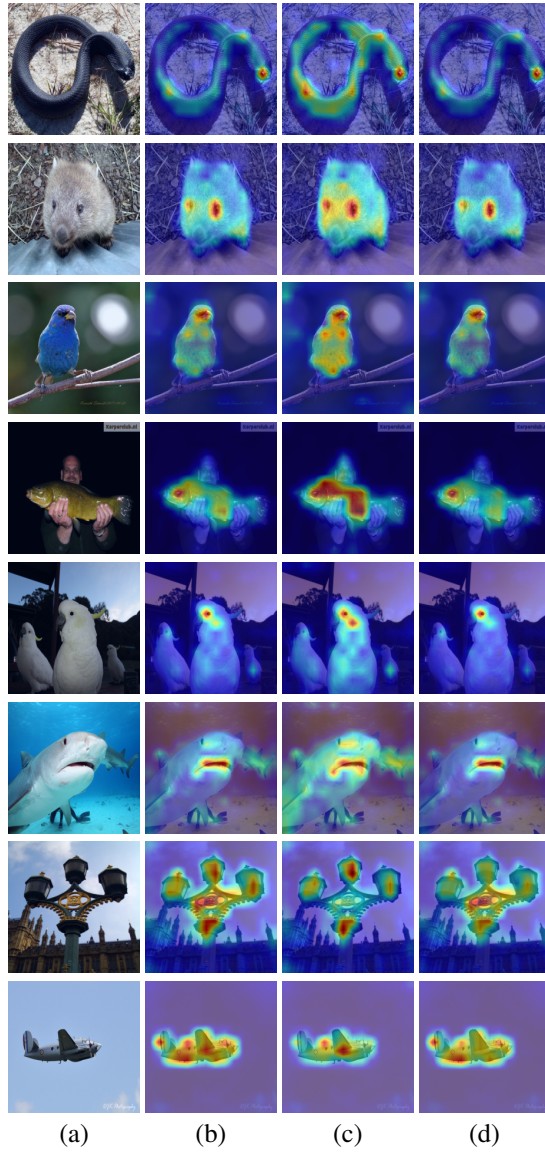

(a)      (b)      (c)      (d)

Figure 3: Results of different explanation methods for DeiT. (a) Input image. (b) PA-LRP (ours), which includes PE attribution. (c) PE only LRP, (d) AttnLRP [2], which does not attribute relevancy to PE.

Figure 3 presents a comparative visualization of these maps. The results reveal three notable trends. **(i) Effectiveness of PE-associated relevance:** The maps from the ablation variant perform at the same level as the AttnLRP baseline. This finding highlights the strength of our method in identifying important signals that previous works have overlooked, underscoring the importance of our PA-LRP rules. **(ii) The uniqueness of PE-associated relevance:** The attributed signal derived solely from positional-associated relevance captures unique relationships, exhibiting clearer spatial and structural patterns. In particular, relevance is distributed across the entire object, especially in the snake, bird, and shark examples. In contrast, the baseline method, which does not propagate relevance through PEs, produces a sparser pattern that does not focus on the entire object but instead is highly selective to specific patches. One possible explanation is that positional-associated relevance better captures concepts related to position, structure, order, and broader regions within the image. **(iii) The importance of balancing:** It is evident that the maps obtained from the PE-associated method and the baseline are complementary, and their combination, extracted via our approach, provides the most robust explanations.

**Quantitative Analysis.** Here, we present our quantitative results through perturbation and segmentation tests.

**Perturbation Tests in Vision.** The results for perturbation tests are shown in Table 1, where we compare our method against the attention LRP baseline. Experiments are conducted using three model sizes: Tiny, Small, and Base.

Notably, our method outperforms the baseline by a significant margin. For instance, in negative perturbation of the predicted class, our method improves the performance by an average of 3.97 points across the three model sizes. However, in positive perturbation, our method lags behind the baseline in half of the cases, though by a small margin of at most 1.2 points.

Table 1: Perturbation Tests for DeiT Variants on ImageNet. AUC results for predicted class. Higher (lower) is better for negative (positive).

| M. Size | Method | Negative ↑ | | Positive ↓ | |
|---------|--------|-----------|--------|-----------|--------|
| | | Predicted | Target | Predicted | Target |
| Base | AttnLRP | 52.185 | 47.516 | 10.784 | **8.032** |
| Base | Ours | **54.970** | **50.174** | **9.918** | 9.237 |
| Small | AttnLRP | 50.662 | 45.105 | 10.511 | 9.761 |
| Small | Ours | **53.482** | **47.948** | **9.135** | **8.477** |
| Tiny | AttnLRP | 43.832 | 37.499 | **2.796** | **2.503** |
| Tiny | Ours | **50.1241** | **42.567** | 3.579 | 3.214 |

**Segmentation Tests in Vision.** As for segmentation tests, the empirical analysis in Tab. 2 clearly demonstrates that our method outperform the AttnLRP [2] baseline. In particular, our method improves over the baseline by 1% points in Pixel Accuracy, and 2% points in mIoU (Mean Intersection over Union). These results further highlight the importance of positionally associated relevance in effectively capturing spatial relationships and representing entire objects more accurately.

Table 2: Segmentation performance of DeiT variants on ImageNet segmentation [20]. Higher is better.

| M. Size | Method | Pixel Acc. ↑ | mIoU ↑ |
|---------|--------|-------------|--------|
| Base | AttnLRP | 72.204 | 50.100 |
| Base | Ours | **72.698** | **51.400** |
| Small | AttnLRP | 72.114 | 50.000 |
| Small | Ours | **73.060** | **51.700** |
| Tiny | AttnLRP | 74.815 | 52.850 |
| Tiny | Ours | **76.613** | **55.920** |

In our experiments, we followed the same guidelines reported as optimal in AttnLRP [2], specifically, a combination of the $\epsilon$-rule and the $\gamma$-rule. We report additional quantitative results in Appendix J, extending our method to the $\alpha$-$\beta$ propagation rule.

## 4.2 Results in NLP

For experiments in the NLP domain, we first present results for perturbation tests, including an ablation study, followed by an assessment of the conservation property. For our tests, we adopt settings defined in [4, 41, 5]. To demonstrate the general superiority of our method beyond LRP-based approaches, we extend our evaluation to additional XAI techniques: Integrated Gradients (IG) [34] and Slalom [24] for classification tasks, and IG [34] and SHAP [18] for zero-shot settings. We note that SHAP is significantly more computationally involved, allowing us to evaluate this method only for smaller context-sized inputs. We present qualitative results in Appendix. D.

Table 3: **Perturbation Tests in NLP.** Evaluation of LLaMa-2 7B and Tiny-LLaMa, finetuned on IMDB, on pruning and generation perturbation tasks. AttnLRP [2] is the LRP baseline. The metrics used are AUAC (area under activation curve, higher is better) and AU-MSE (area under the MSE, lower is better).

| Model | Method | Generation | | Pruning | |
|---|---|---|---|---|---|
| | | AUAC ↑ | AU-MSE ↓ | AUAC ↑ | AU-MSE ↓ |
| LLaMa-2 7B | IG | 0.556 | 24.473 | 0.556 | 24.438 |
| LLaMa-2 7B | Slalom | 0.606 | 18.375 | 0.636 | 7.315 |
| LLaMa-2 7B | AttnLRP | 0.779 | 7.629 | 0.777 | 6.548 |
| LLaMa-2 7B | PE Only | 0.771 | 6.792 | 0.771 | 6.823 |
| LLaMa-2 7B | Ours | **0.796** | **6.521** | **0.790** | **6.325** |
| Tiny-LLaMa-2 7B | IG | 0.637 | 13.745 | 0.636 | 13.770 |
| Tiny-LLaMa-2 7B | Slalom | 0.611 | 15.408 | 0.608 | 15.666 |
| Tiny-LLaMa-2 7B | AttnLRP | 0.803 | 8.065 | 0.792 | 4.030 |
| Tiny-LLaMa-2 7B | PE Only | 0.788 | **3.918** | 0.788 | **3.947** |
| Tiny-LLaMa-2 7B | Ours | **0.806** | 4.915 | **0.805** | 4.082 |

**Perturbation Tests for Finetuned Models.** We conduct perturbation tests on two LLMs, finetuned on the IMDB classification dataset: LLaMa 2-7B [36], and Tiny-LLaMa [40]. The results presented in Table 3 demonstrate that our method achieves better scores than the LRP baseline across all metrics and models. In particular, our approach improves the AU-MSE score in the generation scenario by 14.5% for LLaMa 2-7B and 51.41% for Tiny-LLaMa. To examine the effect of quantization on attributions, we provide additional results for a quantized version of LLaMa 2-7B in Appendix K.

**Perturbation Tests in Zero-Shot Settings** We use LLaMa 3-8B [19] to evaluate explainability performance in zero-shot setting. The results presented in Table 4 showcase the superiority of our method across all metrics. **(i) Multiple-Choice Question Answering (MCQA):** our approach improves, on both generation and pruning scenarios, the AUAC score by approximately 3.2%, and AU-MSE score by approximately by 7.7%. **(ii) Next Token Prediction:** our approach improves the AUAC score by approximately 0.5% on both generation and pruning scenarios, and AU-MSE score by approximately by 3% on both scenarios. In contrast to MCQA, the Wikipedia dataset consists relatively long texts, making shifts in relevancy distributions less critical to the model's prediction.

**Ablation.** We conduct perturbation tests for the method that attributes solely positional-associated relevance. The results are presented in the second, fourth, and sixth rows of Table 3, and second row of Table 5. Surprisingly, this method produces results similar to the AttnLRP baseline, demonstrating the importance of PE-associated relevance, which carries a significant part of the signal. In particular, this variant achieves the best score on the AU-MSE metric for Tiny-LLaMA, reducing the error by 50% compared to AttnLRP [2]. Moreover, in Table 4, we ablate the contribution of our multi-sink approach, which is designed to prevent the loss of positional relevance. We evaluate explainability performance for binary classification of LLaMa-2-7B, using the same perturbation metrics, and report that the multi-sink approach improves the results by 7%.

Table 4: **Ablation Study:** Analyzing the contribution of the multi-sink mechanism via perturbation tests in NLP. The evaluation was conducted on LLaMa-2-7B using the IMDB dataset.

| Method | Generation | | Pruning | |
|---|---|---|---|---|
| | AUAC ↑ | AU-MSE ↓ | AUAC ↑ | AU-MSE ↓ |
| Ours | **0.796** | **6.521** | **0.790** | **6.325** |
| w/o Multi-Sink | 0.759 | 7.124 | 0.758 | 7.158 |

Table 5: **Perturbation Tests in NLP (Zero-Shot).** Evaluation of LLaMa-3 8B in zero-shot on generation and pruning perturbation tasks for both multiple-choice question answering and Next-Token Prediction (NTP) settings. Metrics reported are AUAC (area under activation curve, higher is better) and AU-MSE (area under MSE, lower is better). "AttnLRP" refers to the LRP baseline [2]. 'G' for generation and 'P' for pruning.

| | Multiple-Choice Question Answering | | | | Next Token Prediction | | | |
|---|---|---|---|---|---|---|---|---|
| Method | G. AUAC ↑ | G. AU-MSE ↓ | P. AUAC ↑ | P. AU-MSE ↓ | G. AUAC ↑ | G. AU-MSE ↓ | P. AUAC ↑ | P. AU-MSE ↓ |
| SHAP | 0.291 | 141.757 | 0.282 | 147.229 | - | - | - | - |
| IG | 0.351 | 119.984 | 0.339 | 123.212 | 0.481 | 40.681 | 0.481 | 40.750 |
| AttnLRP | 0.365 | 66.399 | 0.354 | 68.856 | 0.559 | 41.704 | 0.559 | 42.003 |
| PE Only | 0.374 | **61.014** | 0.364 | **63.141** | 0.557 | 40.538 | 0.556 | 40.800 |
| Ours | **0.377** | 61.285 | **0.368** | 63.424 | **0.562** | **40.474** | **0.561** | **40.735** |

## 5   Discussion: The Role of Attributing PEs

Our theoretical and empirical analysis suggests that both semantic and positional relevance are complementary, and combining them is essential to provide precise explanations. LRP-type attribution creates pixel-level heatmaps, but can we characterize and identify which *concepts* are attributed mainly by positional relevance versus semantic relevance?

We may expect, for example, that objects that are usually placed in specific contexts (boats on water, airplanes in the sky) would display a more significant PE component. Much of this position context is relative. RoPE, for example, captures relative position through the matrix multiplication of two position-dependent rotation matrices, which plays a fundamental role in capturing spatial features in vision tasks (e.g., objects spanning across multiple patches) and when modeling relationships between words in the same sentence in NLP. In such cases, our PA-LRP rules can effectively attribute positional features, that are largely ignored by standard LRP methods.

## 6   Conclusions

This paper explores the importance of assigning LRP scores to positional information, a crucial component of Transformers and LLMs. Our theoretical and empirical analysis demonstrates that positional-associated relevance carries a unique type of significance and can drastically improve XAI methods for attention models.

Regarding limitations, we emphasize that our work focuses on designing new custom LRP rules to propagate relevance through PEs, leveraging the insight that this aspect has been previously overlooked. However, we do not extend this insight to redesign or systematically revisit existing LRP rules. Such a redesign could offer an opportunity to empirically and theoretically establish improved LRP rules for attention mechanisms and Transformer models.

## 7   Acknowledgments

This work was supported by a grant from the Tel Aviv University Center for AI and Data Science (TAD). This research was also supported by the Ministry of Innovation, Science & Technology ,Israel (1001576154) and the Michael J. Fox Foundation (MJFF-022407).

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

## A   Background for Additional PEs

In this appendix, we introduce additional PEs beyond those presented in Section 2.

**Relative Positional Bias (RPB).**   Similar to Alibi, RPB [27] modifies the attention scores by introducing a learnable bias term that depends on the relative distance between query and key tokens. For a query at position $i$ and a key at position $j$, the attention scores are adjusted as follows:

$$A_{i,j} = A_{i,j} + B(|i-j|) \tag{10}$$

where $B(i-j)$ is a learned bias function that depends only on the relative position difference $(i-j)$, rather than the absolute positions.

**Attention with Linear Biases (ALiBi).**   ALiBi [26] is a positional encoding method designed to help transformers generalize to longer sequences when trained on shorter ones. Instead of using explicit positional embeddings, ALiBi modifies attention scores directly by introducing a learned linear bias that penalizes attention weights based on token distance.

Specifically, for a query token at position $j$, Alibi adjusts the attention scores as follows:

$$A'_{i,j} = A_{i,j} + m(|i-j|) \tag{11}$$

where $m$ is a learned or predefined slope that controls how quickly attention strength decays with distance. Different attention heads can use different slopes, enabling some heads to focus more on local interactions while others capture long-range dependencies.

## B   PA-LRP Rules for Alibi

Recall the main modification in the ALiBi computation:

$$A'_{i,j} = A_{i,j} + P_{i,j}, \text{ where } P_{i,j} = m(|i-j|) \tag{12}$$

Adopting the same approach presented for RoPE, given the relevancy scores of $A'_{i,j}$, denoted by $\mathcal{R}(A'_{i,j})$, we define specialized rules to propagate relevancy from $A'_{i,j}$ to the positional terms of ALiBi at each layer, namely, indices $i$ and $j$. We begin by distributing the relevancy scores between $A_{i,j}$ and $P_{i,j}$, using the standard $\epsilon$-rule for addition, giving us:

$$\mathcal{R}(P_{i,j}) = P_{i,j} \frac{\mathcal{R}(A'_{i,j})}{A_{i,j} + P_{i,j} + \epsilon} \tag{13}$$

We proceed to propagate the relevancy scores $\mathcal{R}(P_{i,j})$ to the positional encoding $i$ and $j$ in a similar fashion to our rules for RoPE. We make the following observations: (i) $m$ is a constant, resulting in 100% of the relevancy to propagate from $P_{i,j}$ to $|i-j|$. (ii) Since we are using auto-regressive models, we get that $i > j$, allowing us to ignore the absolute value function (iii) The standard $\epsilon$-rule for addition applies the same of subtraction, as we can express $i-j$ as $i+(-j)$, and also $-j = (-1) \cdot j$, and since $-1$ is constant, we propagate the entire relevancy to $j$. That gives us:

$$\mathcal{R}(i) = i \frac{\mathcal{R}(P_{i,j})}{i+(-j)+\epsilon}, \quad \mathcal{R}(j) = \mathcal{R}(-j) = j \frac{\mathcal{R}(P_{i,j})}{i+(-j)+\epsilon} \tag{14}$$

From hereon we adhere to our PA-LRP rules, aggregating the relevance scores of the positional terms across all layers as employed in Section 3.3.

## C   Visualizations - Images

In addition to Figure 3, we provide more examples in Figures 4 - 6. As previously explained, PE-associated relevance better highlights the entire object, and overcomes the issue of over-consideration of the foreground, where extremely high relevancy scores are produced for patches which are more concerned with semantics or common patterns, like a bird's beak in the first row in Figure 5.

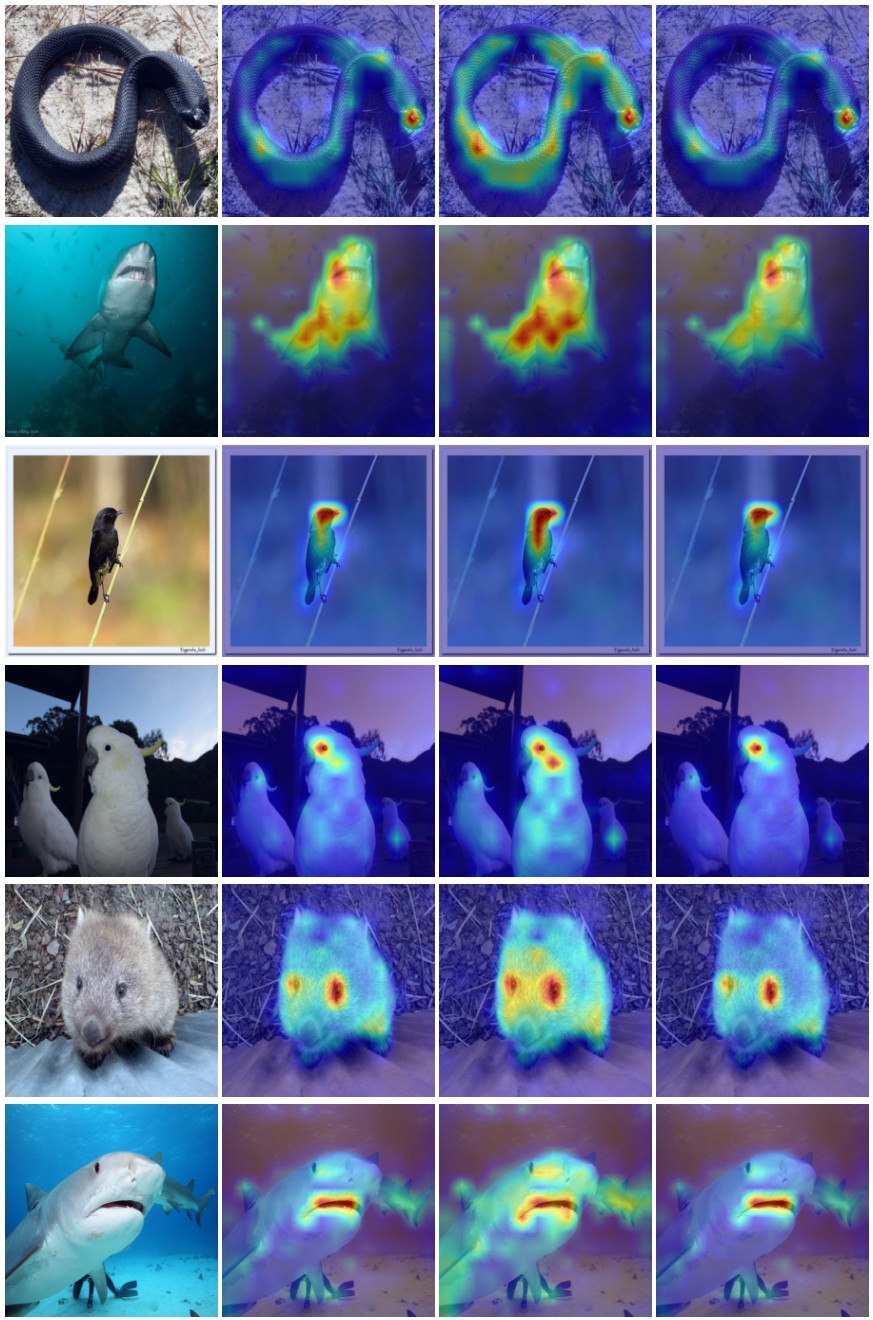

Figure 4: **Additional Qualitative Results In Vision.** Results of different explanation methods for DeiT. (a) The input image. (b) PA-LRP (ours), which include PE relevancy attribution. (c) PE only LRP, (d) AttnLRP [2], which does not attribute relevancy to PE.

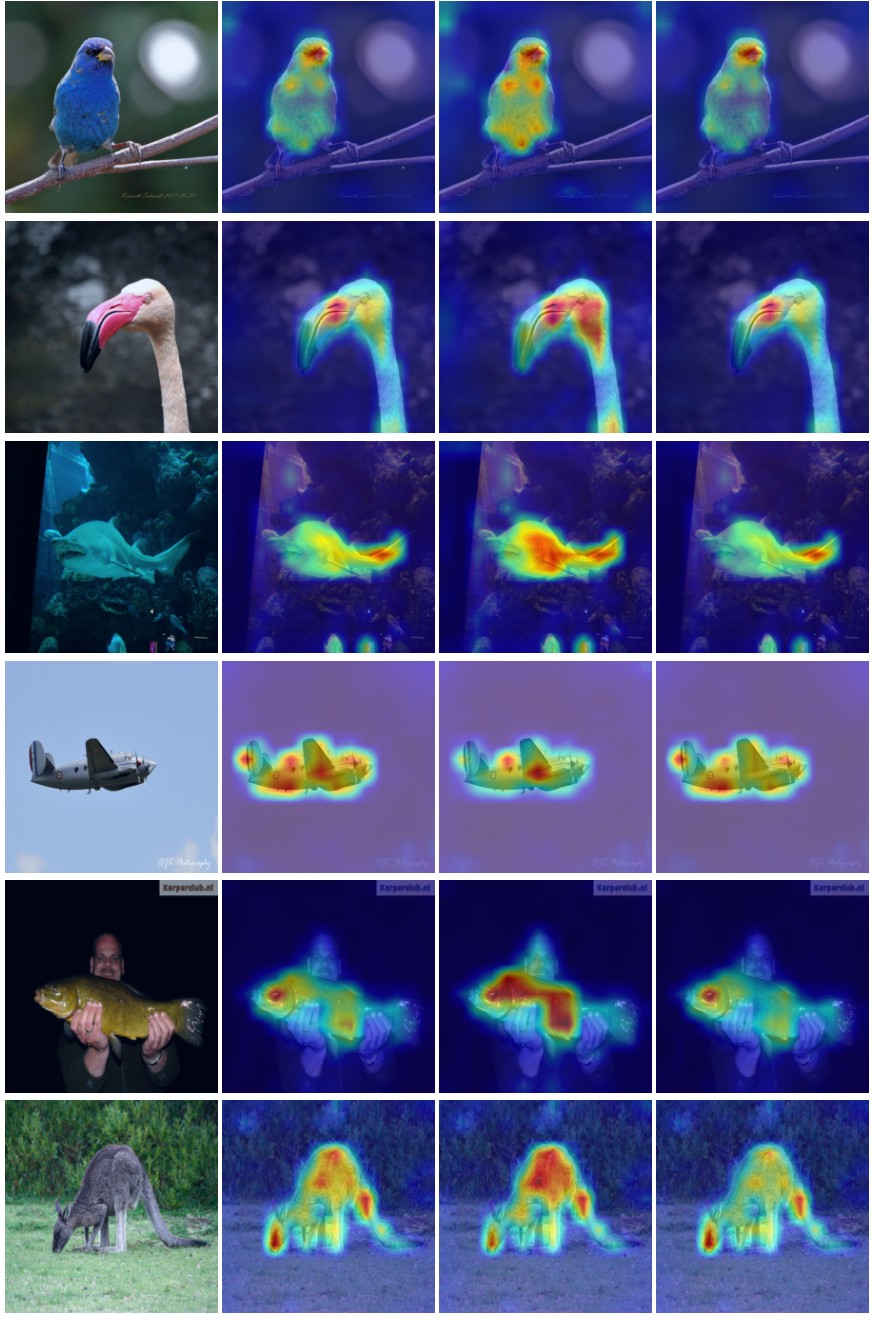

Figure 5: **Additional Qualitative Results In Vision.** Results of different explanation methods for DeiT. (a) The input image. (b) PA-LRP (ours), which include PE relevancy attribution. (c) PE only LRP, (d) AttnLRP [2], which does not attribute relevancy to PE.

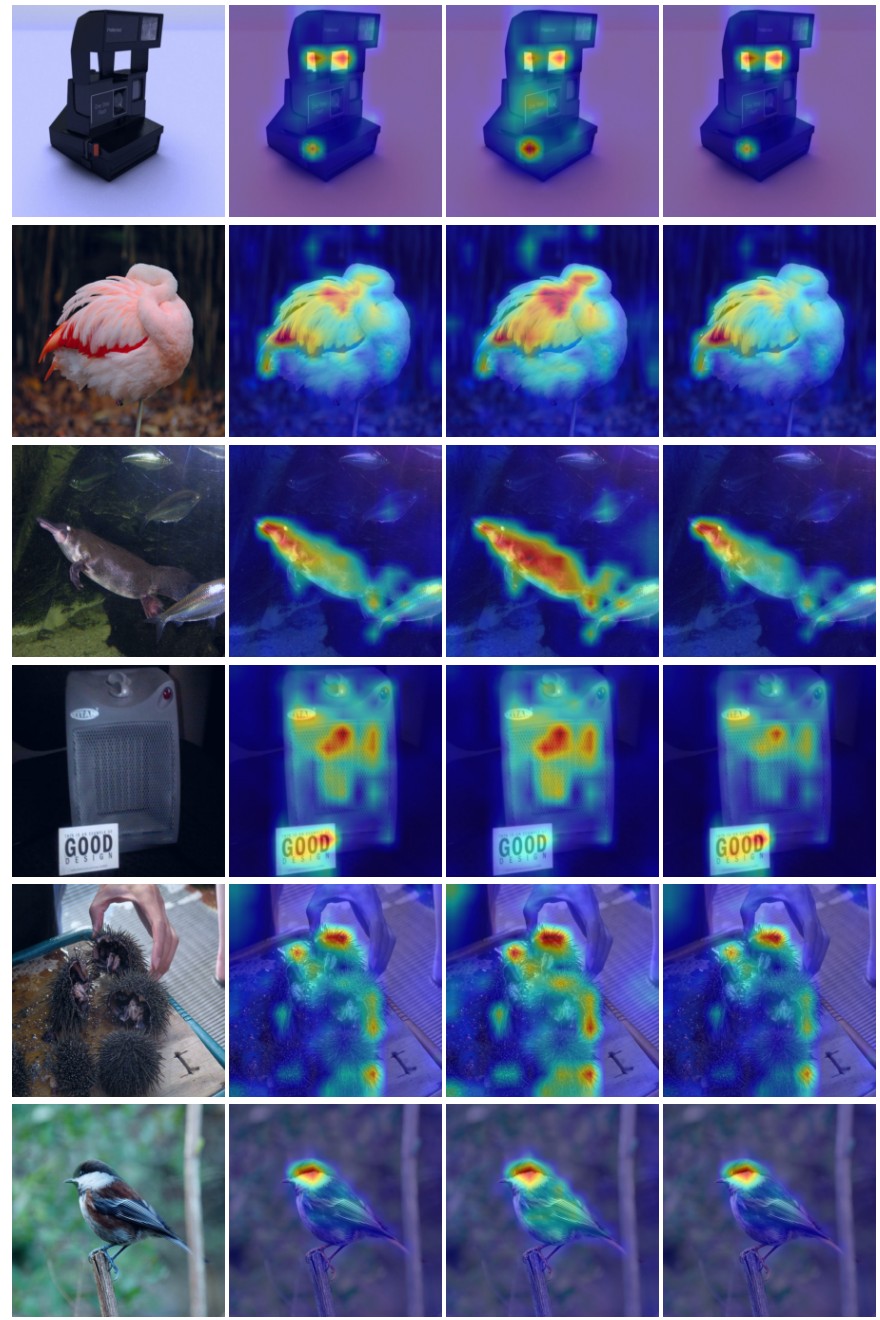

Figure 6: **Additional Qualitative Results In Vision.** Results of different explanation methods for DeiT. (a) The input image. (b) PA-LRP (ours), which include PE relevancy attribution. (c) PE only LRP, (d) AttnLRP [2], which does not attribute relevancy to PE.

# D  Visualizations - Text

We present qualitative results for NLP in Figure 7. It can be seen that our method demonstrates better results in highlighting tokens crucial for prediction, along with their surrounding context, emphasizing it's superiority to draw relevancy based on both semantics and positionally. In (b) we see that the amount of artifacts is reduced drastically, with more relevancy channeled to the tokens essential for prediction ("They should have been giving a tribute to Branagh for bringing us one of the greatest films of all time").

 Great just great ! The West Coast got " Dir ty " Harry Cal la han , the East Coast got Sh ark y . B urt Reyn olds plays Sh ark y in " Sh ark y ' s Machine " and I enjoyed every minute of it . Play ing a ma ver ick n arc ot ics cop in Atlanta , G A is just what everyone wants . Instead of susp ension , he ' s sent to vice squad . Like in the D irty Harry mov ies or any other cop mov ies , the captain is always going to be the j erk . When I was a kid , I was curious what that movie meant " Sh ark y ' s Machine ". Well I knew who played Sh ark y , I wonder what his machine was . It was his GROUP of fellow c ops . After un cover ing the murder , he goes all out to find the per p . When it turns out to be a big time mob b oss , Sh ark y doesn ' t play around . When he gets the other prost itute into safety , Sh ark y f ights back hard and good despite losing a finger to the th ug . And I also like the part where the bad gets blow n out of the building through a plate glass window . That was the B OM B ! R andy C raw ford ' s " St reet Life " really put the movie in the right m ood , and the movie itself is really a great hit to me , AL WA YS ! R ating 4 out of 5 stars .

 Great just great ! The West Coast got " Dir ty " Harry Cal la han , the East Coast got Sh ark y . B urt Reyn olds plays Sh ark y in " Sh ark y ' s Machine " and I enjoyed every minute of it . Play ing a ma ver ick n arc ot ics cop in Atlanta , G A is just what everyone wants . Instead of susp ension , he ' s sent to vice squad . Like in the D irty Harry mov ies or any other cop mov ies , the captain is always going to be the j erk . When I was a kid , I was curious what that movie meant " Sh ark y ' s Machine ". Well I knew who played Sh ark y , I wonder what his machine was . It was his GROUP of fellow c ops . After un cover ing the murder , he goes all out to find the per p . When it turns out to be a big time mob b oss , Sh ark y doesn ' t play around . When he gets the other prost itute into safety , Sh ark y f ights back hard and good despite losing a finger to the th ug . And I also like the part where the bad gets blow n out of the building through a plate glass window . That was the B OM B ! R andy C raw ford ' s " St reet Life " really put the movie in the right m ood , and the movie itself is really a great hit to me , AL WA YS ! R ating 4 out of 5 stars .

 Great just great ! The West Coast got " Dir ty " Harry Cal la han , the East Coast got Sh ark y . B urt Reyn olds plays Sh ark y in " Sh ark y ' s Machine " and I enjoyed every minute of it . Play ing a ma ver ick n arc ot ics cop in Atlanta , G A is just what everyone wants . Instead of susp ension , he ' s sent to vice squad . Like in the D irty Harry mov ies or any other cop mov ies , the captain is always going to be the j erk . When I was a kid , I was curious what that movie meant " Sh ark y ' s Machine ". Well I knew who played Sh ark y , I wonder what his machine was . It was his GROUP of fellow c ops . After un cover ing the murder , he goes all out to find the per p . When it turns out to be a big time mob b oss , Sh ark y doesn ' t play around . When he gets the other prost itute into safety , Sh ark y f ights back hard and good despite losing a finger to the th ug . And I also like the part where the bad gets blow n out of the building through a plate glass window . That was the B OM B ! R andy C raw ford ' s " St reet Life " really put the movie in the right m ood , and the movie itself is really a great hit to me , AL WA YS ! R ating 4 out of 5 stars .

(a)

 I went to see Ham let because I was in between jobs . I figured 4 hours would be great , I ' ve been a fan of Bran agh ; Dead Again , Henry V . I was completely over wh el med by the direction , acting , cinemat ography that this film captured . Like other reviews the 4 hours passes swift ly . Bran agh doesn ' t play Ham let , he is Ham let , he was born for this . When I watch this film I ' m constantly trying to find fault s , I ' ve looked at the go of s and haven ' t noticed them . How he was able to move the camera in and out of the Hall with all the mirror s is a mystery to me . This movie was shot in 7 0 mil . It ' s a shame that Columbia hasn ' t released a W ides creen version of this on V HS . I own a DVD player , and I ' d take this over T itan ic any day . So Columbia if you ' re listening put this film out the way it should be watched ! And I don ' t know what happened at the O sc ars . This should have swe pt Best Picture , Best A ctor , Best D irection , best cinemat ography . What films were they watching ? I felt sorry for Bran agh at the O sc ars when he did a t ribute to Shakespeare on the screen . They should have been giving a t ribute to Bran agh for bringing us one of the greatest films of all time .

 I went to see Ham let because I was in between jobs . I figured 4 hours would be great , I ' ve been a fan of Bran agh ; Dead Again , Henry V . I was completely over wh el med by the direction , acting , cinemat ography that this film captured . Like other reviews the 4 hours passes swift ly . Bran agh doesn ' t play Ham let , he is Ham let , he was born for this . When I watch this film I ' m constantly trying to find fault s , I ' ve looked at the go of s and haven ' t noticed them . How he was able to move the camera in and out of the Hall with all the mirror s is a mystery to me . This movie was shot in 7 0 mil . It ' s a shame that Columbia hasn ' t released a W ides creen version of this on V HS . I own a DVD player , and I ' d take this over T itan ic any day . So Columbia if you ' re listening put this film out the way it should be watched ! And I don ' t know what happened at the O sc ars . This should have swe pt Best Picture , Best A ctor , Best D irection , best cinemat ography . What films were they watching ? I felt sorry for Bran agh at the O sc ars when he did a t ribute to Shakespeare on the screen . They should have been giving a t ribute to Bran agh for bringing us one of the greatest films of all time .

 I went to see Ham let because I was in between jobs . I figured 4 hours would be great , I ' ve been a fan of Bran agh ; Dead Again , Henry V . I was completely over wh el med by the direction , acting , cinemat ography that this film captured . Like other reviews the 4 hours passes swift ly . Bran agh doesn ' t play Ham let , he is Ham let , he was born for this . When I watch this film I ' m constantly trying to find fault s , I ' ve looked at the go of s and haven ' t noticed them . How he was able to move the camera in and out of the Hall with all the mirror s is a mystery to me . This movie was shot in 7 0 mil . It ' s a shame that Columbia hasn ' t released a W ides creen version of this on V HS . I own a DVD player , and I ' d take this over T itan ic any day . So Columbia if you ' re listening put this film out the way it should be watched ! And I don ' t know what happened at the O sc ars . This should have swe pt Best Picture , Best A ctor , Best D irection , best cinemat ography . What films were they watching ? I felt sorry for Bran agh at the O sc ars when he did a t ribute to Shakespeare on the screen . They should have been giving a t ribute to Bran agh for bringing us one of the greatest films of all time .

(b)

Figure 7: **Qualitative Results in NLP.** Both groups (a) and (b) present results from different explanation methods for the same example obtained from the IMDB benchmark. In each group, the first row represents the AttnLRP baseline, followed by the PE-only variant in the middle, and finally, our maps at the end.

# E  Proofs of Lemmas

**Lemma 3.1.** *For input-level PE transformers, the conservation property is violated when disregarding the positional embeddings' relevancy scores.*

*Proof of Lemma 3.1.* Let $Z$ be our input representation to the first transformer layer, such that $Z = P + E$, where $P$ and $E$ are the token and positional embeddings, respectively. Let $L$ be the number of layers in our transformer. Following the conservation property, the sum of the relevancy scores at any given layer $l$ should uphold:

$$\sum \mathcal{R}^{(L)} = \sum \mathcal{R}^{(l)} = \sum \mathcal{R}^{(0)} = \sum R_Z = \sum (\mathcal{R}_E + \mathcal{R}_P) \tag{15}$$

When ignoring $\mathcal{R}_P$, we get the final relevancy attribution map $\mathcal{R}_{input}$, such that:

$$\sum \mathcal{R}^{(l)} = \sum (\mathcal{R}_E + \mathcal{R}_P) \neq \mathcal{R}_E = \mathcal{R}_{input} \tag{16}$$

directly violating the conservation property rule  □

**Lemma 3.2.** *For attention-level PE transformers, our PE-LRP rules satisfy the conservation property.*

*Proof of Lemma 3.2.* Let $M$ be the number of layers in our Transformer, and $L$ the sequence length. We denote $\mathcal{R}^{(l)}$ as the relevancy score of the output at layer $l$. Beginning with $\mathcal{R}^{(M)}$ as the the the model's output propagating relevancy backwards to achieve the final explanation map for the input embeddings $R_E$, we assume that the standard LRP method does not violate conservation, i.e:

$$\forall l \in [M]: \quad \mathcal{R}^{(M)} = \mathcal{R}^{(l)} = \mathcal{R}_E \tag{17}$$

Recall that for our PE-LRP formulation, we achieve the final explanation map by summing together the semantic attribution $\mathcal{R}_E$, achieved by the standard LRP rules, and the positional relevancy $\mathcal{R}_P^{(l)}$ distributed across the absorbing sinks at each attention layer $l \in [M]$, giving us the final relevancy map $R_E + \sum_l \mathcal{R}_P^{(l)}$. We aim to prove the following:

$$\mathcal{R}^{(M)} = \mathcal{R}_E + \sum_l \mathcal{R}_P^{(l)} \tag{18}$$

Each attention layer in the transformer is computed using rotary attention:

$$\forall j \in [L] : \tilde{\boldsymbol{Q}}_j = R_j \boldsymbol{Q}_j, \quad \tilde{\boldsymbol{K}}_j = R_j \boldsymbol{K}_j, \quad \text{Rotary Attention}(X) = \text{Softmax}\left(\frac{\tilde{\boldsymbol{Q}}\tilde{\boldsymbol{K}}^T}{\sqrt{d_k}}\right) V. \tag{19}$$

Notice that any computation in this layer which involves more than one tensor, is a matrix multiplication function. Adopting the existing baseline, we use the uniform relevance propagation rule, distributing the relevancy evenly between components. Thus, the relevancy scores of $Q, K, V, P$, with $P$ denoting the rotation matrix, is equal, and added together to the relevancy of the attention layer's output. The absorbing sink mechanism results in the following:

$$\mathcal{R}^{(0)} = \mathcal{R}_E, \quad \forall l \in [M]: \quad \mathcal{R}^{(l)} = \mathcal{R}^{(l-1)} + \mathcal{R}_P^{(l)} \tag{20}$$

Following this recursion we would get the exact same result as Eq. 18  □

**Lemma 3.3.** *For attention-level PE transformers, current LRP attribution rules achieve low faithfulness, especially when considering positional features.*

*Proof of Lemma 3.3.* We define a basic learning problem which relies solely on positional features, proving that existing LRP-based explanation methods which don't propagate relevance through positional encodings, will not produce faithful explanations. Let us assume we use an auto-regressive transformer model (e.g GPT), with a single causal self-attention with Alibi PE, and the Value projection replaced by an affine transformation (instead of a linear layer). Also, for brevity, let us consider scalar input tokens with sequence length of $L = 2$, denoted by $x_1, x_2$. The final model uses the following keys ($K$), queries ($Q$), and values ($V$):

$$\forall i \in [1, 2]: \quad Q_i = W_Q X_i, \quad , K_i = W_K X_i, \quad V_i = W_V X_i + b \tag{21}$$

We apply the Alibi self-attention mechanism, and obtain the final output $O = (O_1, O_2)$:

$$A(i, j) = \frac{Q_i K_j}{\sqrt{d}} + m_h(i - j), \quad m_h = 1, \quad O_2 = A_{2,1} V_1 + A_{2,2} V_2 \tag{22}$$

To prevent the semantic representation from affecting the prediction, an optimal solution to this problem will assign zeros to $W_Q, W_K$, namely: $Q = K = \begin{pmatrix} 0 \\ 0 \end{pmatrix}$. For the Value projection, we assume: $V = W_v + b$, with $W_v = \begin{pmatrix} 0 \\ 0 \end{pmatrix}, b \neq \begin{pmatrix} 0 \\ 0 \end{pmatrix}$.

**Relevance propagation.** Following our settings, we get:

$$A = \begin{pmatrix} 0 & 0 \\ 1 & 0 \end{pmatrix}, \text{ giving us: } Attention(Q, K, V) = A \times V = \begin{pmatrix} 0 & 0 \\ 1 & 0 \end{pmatrix} b \tag{23}$$

For the backwards relevancy propagation, the relevancy scores of $Attention$ are distrusted between $A$ and $V$ based on the standard Gradient $\times$ Input. Regardless, we now consider how relevancy scores of both terms $\mathcal{R}_V, \mathcal{R}_{score}$ are propagated back to the input $x$.

- $\mathcal{R}_V \rightarrow \mathcal{R}_x$. recall that $W_v$ are assigned with zeros. Given that the fundamental $\epsilon$-LRP rule for affine transformations ignores the bias term completely and uses the weights $W_v$ as a measure of weighting the relevancy scores, we get that zero relevancy scores are produced for both tokens.

- $\mathcal{R}_A \rightarrow \mathcal{R}_x$. Following the standard LRP rules, the positional terms would be considered a constant, and therefor, 100% of the distribution would be directed to the queries and keys. Given that $W_Q, W_K$ are assigned with zeros, we again get zero relevancy scores being propagated to $x$.

Given that the relevancy scores propagated back from the attention layer are all assigned with zeros, we will get a final attribution map of zeroes, indicating the same level of impact for all tokens. This of course, yields an unfaithful explanation. In contrast, our method makes positional terms attributable, maintaining relevancy scores that would otherwise be zeroed out due to existing rules. □

## F Conservation Percentage Results

We measure the sum of relevance for DeiT model at different capacities: Tiny, Small, and Base. The figure provides clear visualization for the violation of the conservation property, with PE relevancy constituting 16.75%, 22.39%, and 9.22% out of the total relevancy for Tiny, Small, and Base DeiT models, respectively.

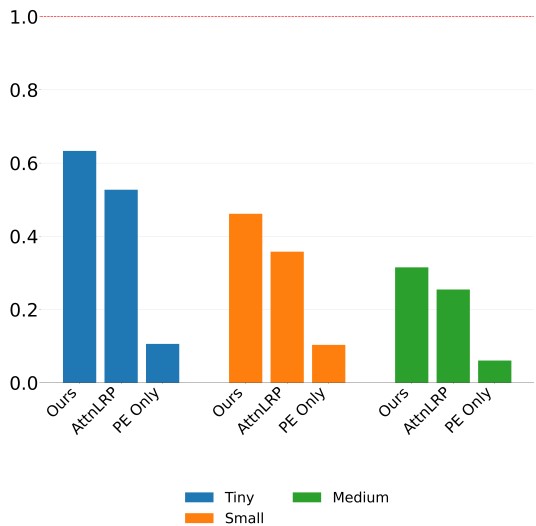

Figure 8: We assess both the positional relevance and the non-positional relevance for DeiT models at different capacities, visualizing the violation of conservation rule, with high non-negligible ratio between the entire relevance in the models 'ours' and positional-associated relevance 'PE Only'.

## G Experiments Compute Resources

All experiments were conducted using the PyTorch framework on publicly available datasets and executed on a single NVIDIA H100-80GB HBM3 GPU, running for at most 12 hours.

## H Experimental Setup

PyTorch is used for conducting all of our experiments. We note that while exploring optimal rules for PE-LRP through theoretical analysis or grid search could further improve our empirical results, we deliberately chose not to perform hyperparameter tuning for our PE-LRP rules in order to neutralize the impact of hyperparameter search.

We provide the number of randomly chosen samples used for perturbation evaluation, for each dataset:

- Imagenet: 12,500 samples
- ARC-Easy : 5,200 samples
- Wiki-text: 5,000
- IMDB: 5,000

# I  Statistical Analysis

We have added comprehensive statistical analyses, including variance measures and paired t-tests, for the empirical experiments. For the paired t-tests, we compare each baseline with our method, where each paired comparison is performed on the classification accuracy for each sample across all perturbation steps. We report the full results in Tables 6–8.

Table 6: **Perturbation Tests in NLP.** Evaluation of LLaMa-2 7B and Tiny-LLaMa, finetuned on IMDB, on pruning and generation perturbation tasks. AttnLRP [2] is the LRP baseline. The metrics used are AUAC (area under activation curve, higher is better) and AU-MSE (area under the MSE, lower is better).

| Model | Method | Generation | | Pruning | |
|---|---|---|---|---|---|
| | | AUAC ↑ | AU-MSE ↓ | AUAC ↑ | AU-MSE ↓ |
| LLaMa-2 7B | IG | $0.556 \pm 0.03$ (P:0.00,T:24.28) | $24.473 \pm 0.30$ (P:0.00,T:-37.49) | $0.556 \pm 0.03$ (P:0.00,T:24.46) | $24.438 \pm 0.30$ (P:0.00,T:24.46) |
| LLaMa-2 7B | Slalom | $0.606 \pm 0.03$ (P:0.00,T:20.91) | $18.375 \pm 0.28$ (P:0.00,T:-28.64) | $0.636 \pm 0.03$ (P:0.00,T:17.82) | $7.315 \pm 0.27$ (P:0.00,T:-27.63) |
| LLaMa-2 7B | AttnLRP | $0.779 \pm 0.05$ (P:0.02,T:2.29) | $7.629 \pm 0.28$ (P:0.00,T:-13.31) | $0.777 \pm 0.05$ (P:0.00,T:6.47) | $6.548 \pm 0.30$ (P:0.00,T:-12.18) |
| LLaMa-2 7B | PE Only | $0.771 \pm 0.03$ | $6.792 \pm 0.20$ | $0.771 \pm 0.03$ | $6.823 \pm 0.20$ |
| LLaMa-2 7B | Ours | $\mathbf{0.796 \pm 0.03}$ | $\mathbf{6.521 \pm 0.19}$ | $\mathbf{0.790 \pm 0.03}$ | $\mathbf{6.325 \pm 0.19}$ |
| Tiny-LLaMa-2 7B | IG | $0.637 \pm 0.03$ (P:0.00,T:29.25) | $13.745 \pm 0.24$ (P:0.00,T:-33.88) | $0.636 \pm 0.03$ (P:0.00,T:29.98) | $13.770 \pm 0.24$ (P:0.00,T:-34.54) |
| Tiny-LLaMa-2 7B | Slalom | $0.611 \pm 0.03$ (P:0.00,T:22.03) | $15.408 \pm 0.26$ (P:0.00,T:-30.86) | $0.608 \pm 0.03$ (P:0.00,T:20.67) | $15.666 \pm 0.26$ (P:0.00,T:-31.20) |
| Tiny-LLaMa-2 7B | AttnLRP | $0.803 \pm 0.05$ (P:0.00,T:0.57) | $8.065 \pm 0.26$ (P:0.00,T:-29.47) | $0.792 \pm 0.05$ (P:0.01,T:2.49) | $4.030 \pm 0.22$ (P:0.00,T:1.1) |
| Tiny-LLaMa-2 7B | PE Only | $0.788 \pm 0.03$ | $\mathbf{3.918 \pm 0.16}$ | $0.788 \pm 0.03$ | $\mathbf{3.947 \pm 0.16}$ |
| Tiny-LLaMa-2 7B | Ours | $\mathbf{0.806 \pm 0.03}$ | $4.915 \pm 0.15$ | $\mathbf{0.805 \pm 0.15}$ | $4.082 \pm 0.05$ |

Table 7: **Perturbation Tests in NLP (Zero-Shot).** Evaluation of LLaMa-3 8B in zero-shot on generation and pruning perturbation tasks for both multiple-choice question answering and Next-Token Prediction (NTP) settings. Metrics reported are AUAC (area under activation curve, higher is better) and AU-MSE (area under MSE, lower is better). "AttnLRP" refers to the LRP baseline [2]. 'G' for generation and 'P' for pruning.

| Method | Multiple-Choice Question Answering | | | | Next Token Prediction | | | |
|---|---|---|---|---|---|---|---|---|
| | G. AUAC ↑ | G. AU-MSE ↓ | P. AUAC ↑ | P. AU-MSE ↓ | G. AUAC ↑ | G. AU-MSE ↓ | P. AUAC ↑ | P. AU-MSE ↓ |
| SHAP | $0.291 \pm 0.04$ | $141.757 \pm 0.71$ | $0.282 \pm 0.04$ | $147.229 \pm 0.71$ | - | - | - | - |
| IG | $0.351 \pm 0.02$ (P:0.00,T:4.46) | $119.984 \pm 0.50$ (P:0.00,T:-64.09) | $0.339 \pm 0.02$ (P:0.00,T:4.22) | $123.212 \pm 0.50$ (P:0.00,T:-63.63) | $0.481 \pm 0.04$ (P:0.00,T:16.45) | $40.681 \pm 0.59$ (P:0.78,T:-0.05) | $0.481 \pm 0.04$ (P:0.00,T:16.46) | $40.750 \pm 0.58$ (P:0.96,T:-0.05) |
| AttnLRP | $0.365 \pm 0.04$ (P:0.00,T:4.84) | $66.399 \pm 0.56$ (P:0.00,T:-9.79) | $0.354 \pm 0.04$ (P:0.00,T:5.43) | $68.856 \pm 0.57$ (P:0.00,T:-10.33) | $0.559 \pm 0.04$ (P:0.06,T:1.90) | $41.704 \pm 0.58$ (P:0.00,T:-21.24) | $0.559 \pm 0.04$ (P:0.06,T:1.87) | $42.003 \pm 0.58$ (P:0.00,T:-21.47) |
| PE Only | $0.374 \pm 0.04$ (P:0.01,T:2.51) | $\mathbf{61.014 \pm 0.47}$ (P:0.01,T:2.60) | $0.364 \pm 0.04$ (P:0.01,T:2.69) | $\mathbf{63.141 \pm 0.47}$ (P:0.00,T:2.85) | $0.557 \pm 0.04$ (P:0.00,T:11.74) | $40.538 \pm 0.58$ (P:0.00,T:-3.30) | $0.556 \pm 0.04$ (P:0.00,T:11.27) | $40.800 \pm 0.58$ (P:0.00,T:-3.20) |
| Ours | $\mathbf{0.377 \pm 0.04}$ | $61.285 \pm 0.47$ | $\mathbf{0.368 \pm 0.04}$ | $63.424 \pm 0.47$ | $\mathbf{0.562 \pm 0.58}$ | $\mathbf{40.474 \pm 0.34}$ | $\mathbf{0.561 \pm 0.04}$ | $\mathbf{40.735 \pm 0.58}$ |

Table 8: Perturbation Tests for DeiT Variants on ImageNet. AUC results for predicted class. Higher (lower) is better for negative (positive).

| M. Size | Method | Negative ↑ | | Positive ↓ | |
|---|---|---|---|---|---|
| | | Predicted | Target | Predicted | Target |
| Base | AttnLRP | $52.185 \pm 0.03$ (P:0.00,T:20.60) | $47.516 \pm 0.01$ (P:0.00,T:14.63) | $10.784 \pm 0.01$ (P:0.00,T:-9.20) | **$8.032 \pm 0.00$** (P:0.00,T:7.78) |
| Base | Ours | **$54.970 \pm 0.03$** | **$50.174 \pm 0.02$** | **$9.918 \pm 0.03$** | $9.237 \pm 0.02$ |
| Small | AttnLRP | $50.662 \pm 0.03$ (P:0.00,T:22.20) | $45.105 \pm 0.02$ (P:0.00,T:17.73) | $10.511 \pm 0.03$ (P:0.00,T:-16.35) | $9.761 \pm 0.02$ (P:0.00,T:-14.22) |
| Small | Ours | **$53.482 \pm 0.03$** | **$47.948 \pm 0.02$** | **$9.135 \pm 0.03$** | **$8.477 \pm 0.02$** |
| Tiny | AttnLRP | $43.832 \pm 0.03$ (P:0.00,T:53.15) | $37.499 \pm 0.02$ (P:0.00,T:45.70) | **$2.796 \pm 0.03$** (P:0.00,T:22.36) | **$2.503 \pm 0.02$** (P:0.00,T:20.69) |
| Tiny | Ours | **$50.1241 \pm 0.03$** | **$42.567 \pm 0.02$** | $3.579 \pm 0.03$ | $3.214 \pm 0.02$ |

The results demonstrate that all differences reported in the paper are statistically significant, as verified using paired t-tests. We observe $p < 0.03$ figures across all experiments, with most $p$-values lower than $10^{-4}$. The variance is consistently low, and paired t-tests yield small $p$-values, while $t$-scores are aligned with performance trends.

## J Perturbation Test for Vision Transformer: EpslionGamma vs. AlphaBeta

To evaluate the robustness of our method against other LRP-based rules (Tab. 9), we conduct additional perturbation experiments for the vision transformer, comparing the settings adopted by AttnLRP [2], which are considered optimal and utilize a mix between the $\gamma$ and $\epsilon$ rules, with the well-adopted $\alpha$-$\beta$ propagation rule, adopted in various prominent XAI papers (e.g. [14]).

Table 9: Perturbation Tests for DeiT Variants on ImageNet, comparing $\gamma$-$\epsilon$, and $\alpha$-$\beta$ LRP rules, for AttnLRP [2] and our method.

| M. Size | Method | Negative ↑ | | Positive ↓ | |
|---|---|---|---|---|---|
| | | Predicted | Target | Predicted | Target |
| Base | AttnLRP $(\alpha, \beta)$ | 44.612 | 40.542 | **9.172** | 8.515 |
| Base | AttnLRP $(\epsilon, \gamma)$ | **52.185** | **47.516** | 10.784 | **8.032** |
| Base | Ours $(\alpha, \beta)$ | **61.974** | **55.237** | 43.557 | 39.499 |
| Base | Ours $(\epsilon, \gamma)$ | 54.970 | 50.174 | **9.918** | **9.237** |
| Small | AttnLRP $(\alpha, \beta)$ | 45.698 | 40.890 | **9.822** | **9.048** |
| Small | AttnLRP $(\epsilon, \gamma)$ | **50.662** | **45.105** | 10.511 | 9.761 |
| Small | Ours $(\alpha, \beta)$ | **62.963** | **55.114** | 33.651 | 30.158 |
| Small | Ours $(\epsilon, \gamma)$ | 53.482 | 47.948 | **9.135** | **8.477** |
| Tiny | AttnLRP $(\alpha, \beta)$ | 31.297 | 26.962 | 11.301 | 9.891 |
| Tiny | AttnLRP $(\epsilon, \gamma)$ | **43.832** | **37.499** | **2.796** | **2.503** |
| Tiny | Ours $(\alpha, \beta)$ | **56.925** | **46.835** | 29.049 | 24.468 |
| Tiny | Ours $(\epsilon, \gamma)$ | 50.1241 | 42.567 | **3.579** | **3.214** |

The results demonstrate that using the settings presented in AttnLRP [2] $(\epsilon, \gamma)$ provides the optimal performance for both methods. We underline that while the alpha-beta rule manages to achieve higher performance for negative perturbation for our method, it is accompanied by a drastic tradeoff with positive perturbations, which implies that the attribution map is coupled with high level of noise.

## K Perturbation Test for Quantized Large Language Transformer

To evaluate the robustness of our method on quantized models (Tab. 10), we conduct additional perturbation experiments for a quantized version of LLaMa 2-7B , finetuned on the IMDB classification dataset.

Table 10: **Perturbation Tests in NLP.** Evaluation of quantized LLaMa-2 7B, finetuned on IMDB, on pruning and generation perturbation tasks. AttnLRP [2] is the LRP baseline. The metrics used are AUAC (area under activation curve, higher is better) and AU-MSE (area under the MSE, lower is better).

| Model | Method | Generation | | Pruning | |
|---|---|---|---|---|---|
| | | AUAC ↑ | AU-MSE ↓ | AUAC ↑ | AU-MSE ↓ |
| LLaMa-2 7B Quantized | AttnLRP | 0.774 | 11.348 | 0.767 | 10.067 |
| LLaMa-2 7B Quantized | PE Only | 0.758 | 10.730 | 0.758 | 10.774 |
| LLaMa-2 7B Quantized | Ours | **0.785** | **10.137** | **0.778** | **9.685** |

The results presented in Table 10 demonstrate that quantization does not affect the effectiveness of our method, which consistently outperforms the baseline. In particular, our approach improves the AU-MSE score in the generation scenario by 10.6%.

## L Broader Impacts

Our work proposes a novel XAI technique that enhances the explainability of transformer models. XAI plays a critical role in ensuring the safe and responsible deployment of machine learning systems, particularly in high-stakes domains such as healthcare, finance, and law. It helps users and researchers understand, trust, and effectively audit model decisions. In particular, XAI facilitates the detection of biases, identification of failure modes, and debugging of unintended behaviors. By providing more

accurate and faithful explanations for transformer-based models, our method contributes to greater transparency and accountability in DL systems. For all of those reasons, we believe this work will positively impact both the research community and practical applications by enabling safer and more interpretable use of powerful language models.

