# OpenReview forum: "Revisiting LRP: Positional Attribution as the Missing Ingredient for Transformer Explainability"
_NeurIPS.cc/2025/Conference — NeurIPS 2025 poster_

### Official Review · Reviewer_jDLx · 2025-06-29

**Clarity:** 3
**Significance:** 3
**Originality:** 3
**Rating:** 4
**Confidence:** 3

**Summary:**

This paper proposes Positional-Aware Layer-wise Relevance Propagation (PA-LRP), a novel explainability method for Transformers that addresses a key limitation in existing LRP-based approaches—the omission of positional encodings (PE) in attribution. The proposed method differs from prior work in two main ways: (1) it reformulates the input space to incorporate both token and positional information, and (2) it introduces the first LRP rules specifically designed to propagate relevance through standard PE schemes, including Learnable, Sinusoidal, and Attention-level positional encodings such as Rotary PE.

The authors provide a thorough theoretical analysis, proving that existing methods violate the conservation property and exhibit low faithfulness in tasks that depend heavily on positional information. In contrast, the proposed PA-LRP satisfies the conservation principle and enables more faithful attribution.

Extensive experiments across vision and NLP tasks -- including evaluations on DeiT and LLaMA models -- demonstrate that PA-LRP outperforms baseline methods in most settings.

**Questions:**

1. In Figure 3, the explainability heatmaps generated by PA-LRP and AttnLRP appear visually similar. Could you elaborate on the qualitative differences that PA-LRP captures and explain why these differences might not be visually apparent despite the claimed improvements?

2. For the perturbation and segmentation tests (Tables 1–3), how many samples were included in the evaluation? Were the results obtained from the full test set or a selected subset? Did you run multiple trials, and if so, could you report any measures of variance or statistical significance?

**Ethical Concerns:**

["NO or VERY MINOR ethics concerns only"]

**Final Justification:**

I've read the authors' rebuttal, and they have fully addressed my concern regarding Q2 and partially addressed my concern regarding Q1. I will keep my score.

**Limitations:**

yes

**Quality:**

3

**Strengths And Weaknesses:**

Strengths:
1. The authors identify a critical yet previously overlooked limitation in existing LRP-based explainability methods—the lack of attribution to positional encodings, which are essential to Transformer models.

2. The paper introduces new LRP propagation rules tailored for Learnable, Sinusoidal, and Rotary positional encodings, and interprets positional information as a vector passed through attention blocks via a semi-skip connection mechanism, enabling more faithful propagation.

3. The authors provide formal theoretical justifications showing that prior methods violate the conservation property, and prove that the proposed PA-LRP rules satisfy it, ensuring consistency and faithfulness in attribution.

Weakness:
1. The qualitative examples in Figure 3 do not clearly demonstrate the superiority of PA-LRP over the AttnLRP baseline. The visualizations appear very similar, which weakens the visual impact of the proposed method.

2. Although Tables 1, 2, and 3 report improved quantitative results, the experimental setup lacks detail. In particular, it is unclear how many samples were used in the evaluation, how they were selected, and whether any variance or statistical significance was assessed. The absence of error bars or multiple runs raises concerns about the robustness of the reported improvements.

Minor:
The term "multi-sink mechanism" used in Table 4 is not explicitly introduced or defined in the main text. While it appears to refer to the layer-wise aggregation of positional relevance across attention layers (as described in Section 3.1 and 3.3), clarifying this terminology when the concept is first introduced would improve readability and help readers better interpret the ablation results. A brief explanation or subsection referencing this mechanism explicitly would be helpful.

---

> ### Author Rebuttal · Authors · 2025-07-31
>
> Thank you for the comprehensive review and the constructive feedback.
>
>
> >W1. The qualitative examples in Figure 3 do not clearly demonstrate the superiority of PA-LRP over the AttnLRP baseline. The visualizations appear very similar, which weakens the visual impact of the proposed method.
>
> Our most important result is on attention-level PE, such as Rotary PE (RoPE). In this setting, the PE is integrated at each attention layer, making it significantly more impactful in both the forward and backward passes. In this regime, our empirical results are substantially better, as shown by the large performance gap between our method and AttnLRP in Tables 3 and 5, and by the qualitative examples in Figure 7 (for example, note the blue distracting words attributed by AttnLRP, such as “West,” “Harry,” and “Reyn”). Additionally, our theoretical findings are more relevant in this setting. For example, Theorem 3.3 proves that previously proposed LRP methods are not faithful when positional features are involved. It is also important to highlight that attention-level PEs are widely popular in both LLMs (Qwen, LLama, Pythia and others) and vision models like SAM2 (see line 108 in our paper).
>
> >W2 & Q2. Although Tables 1, 2, and 3 report improved quantitative results, the experimental setup lacks detail. In particular, it is unclear how many samples were used in the evaluation, how they were selected, and whether any variance or statistical significance was assessed. The absence of error bars or multiple runs raises concerns about the robustness of the reported improvements…. Q2. For the perturbation and segmentation tests (Tables 1–3), how many samples were included in the evaluation? Were the results obtained from the full test set or a selected subset? Did you run multiple trials, and if so, could you report any measures of variance or statistical signifiance?
>
>
> We thank the reviewer for the suggestion regarding statistical significance, which we address with dedicated experiments. To ensure the robustness of our results, we report variance across runs and conduct paired t-tests to support statistical significance. The results will appear in the camera-ready version, and we add them to all of the relevant experiments (Tables 1-5). Please see Tables A and B below for perturbation tests in NLP. We demonstrate that the variance is consistently low, and paired t-tests yield small p-values with t-scores aligned with performance trends.
>
> **Table A:** Table 3 from our paper (perturbations in NLP under the finetuning regime), with added statistical information. The P-values are obtained using paired t-tests compared to our method.
>
>
> | Model           | Method  | AUAC ↑ (Gen)                  | AU-MSE ↓ (Gen)                  | AUAC ↑ (Prune)                | AU-MSE ↓ (Prune)                |
> | --------------- | ------- | ----------------------------- | ------------------------------- | ----------------------------- | ------------------------------- |
> | LLaMa-2 7B      | IG      | 0.556 ± 0.03 (P:0.00,T:24.28) | 24.473 ± 0.30 (P:0.00,T:-37.49) | 0.556 ± 0.03 (P:0.00,T:24.46) | 24.438 ± 0.30 (P:0.00,T:24.46)  |
> | LLaMa-2 7B      | Slalom  | 0.606 ± 0.03 (P:0.00,T:20.91) | 18.375 ± 0.28 (P:0.00,T:-28.64) | 0.636 ± 0.03 (P:0.00,T:17.82) | 7.315 ± 0.27 (P:0.00,T:-27.63)  |
> | LLaMa-2 7B      | AttnLRP | 0.779 ± 0.05 (P:0.02,T:2.29)  | 7.629 ± 0.28 (P:0.00,T:-13.31)  | 0.777 ± 0.05 (P:0.00,T:6.47)  | 6.548 ± 0.30 (P:0.00,T:-12.18)  |
> | LLaMa-2 7B      | PE Only | 0.771 ± 0.03                  | 6.792 ± 0.20                    | 0.771 ± 0.03                  | 6.823 ± 0.20                    |
> | LLaMa-2 7B      | Ours    | **0.796 ± 0.03**              | **6.521 ± 0.19**                | **0.790 ± 0.03**              | **6.325 ± 0.19**                |
> |||||||
> | Tiny-LLaMa-2 7B | IG      | 0.637 ± 0.03 (P:0.00,T:29.25) | 13.745 ± 0.24 (P:0.00,T:-33.88) | 0.636 ± 0.03 (P:0.00,T:29.98) | 13.770 ± 0.24 (P:0.00,T:-34.54) |
> | Tiny-LLaMa-2 7B | Slalom  | 0.611 ± 0.03 (P:0.00,T:22.03) | 15.408 ± 0.26 (P:0.00,T:-30.86) | 0.608 ± 0.03 (P:0.00,T:20.67) | 15.666 ± 0.26 (P:0.00,T:-31.20) |
> | Tiny-LLaMa-2 7B | AttnLRP | 0.803 ± 0.05 (P:0.00,T:0.57)  | 8.065 ± 0.26 (P:0.00,T:-29.47)  | 0.792 ± 0.05 (P:0.01,T:2.49)  | 4.030 ± 0.22 (P:0.00,T:1.1)     |
> | Tiny-LLaMa-2 7B | PE Only | 0.788 ± 0.03                  | **3.918 ± 0.16**                | 0.788 ± 0.03                  | **3.947 ± 0.16**                |
> | Tiny-LLaMa-2 7B | Ours    | **0.806 ± 0.03**              | 4.915 ± 0.15               | **0.805 ± 0.15**              | 4.082 ± 0.05                    |
>
>
> ___
>
> **Table B:** Table 5 from our paper (perturbations in NLP in zero-shot settings), with added statistical information.
>
> | Method   | G. AUAC ↑           | G. AU-MSE ↓          | P. AUAC ↑           | P. AU-MSE ↓         | G. AUAC ↑           | G. AU-MSE ↓          | P. AUAC ↑           | P. AU-MSE ↓         |
> |----------|---------------------|----------------------|---------------------|---------------------|---------------------|----------------------|---------------------|---------------------|
> |          | **MCQA**  |    **MCQA**  | **MCQA** | **MCQA** | **Token Prediction** | **Token Prediction** | **Token Prediction** | **Token Prediction** |
> | SHAP     | 0.291 ± 0.04        | 141.757 ± 0.71       | 0.282 ± 0.04        | 147.229 ± 0.71      | -                   | -                    | -                   | -                   |
> | IG       | 0.351 ± 0.02 (P:0.00,T:4.46) | 119.984 ± 0.50 (P:0.00,T:-64.09) | 0.339 ± 0.02 (P:0.00,T:4.22) | 123.212 ± 0.50 (P:0.00,T:-63.63) | 0.481 ± 0.04 (P:0.00,T:16.45) | 40.681 ± 0.59 (P:0.78,T:-0.05) | 0.481 ± 0.04 (P:0.00,T:16.46) | 40.750 ± 0.58 (P:0.96,T:-0.05) |
> | AttnLRP  | 0.365 ± 0.04 (P:0.00,T:4.84) | 66.399 ± 0.56 (P:0.00,T:-9.79) | 0.354 ± 0.04 (P:0.00,T:5.43) | 68.856 ± 0.57 (P:0.00,T:-10.33) | 0.559 ± 0.04 (P:0.06,T:1.90) | 41.704 ± 0.58 (P:0.00,T:-21.24) | 0.559 ± 0.04 (P:0.06,T:1.87) | 42.003 ± 0.58 (P:0.00,T:-21.47) |
> | PE Only  | 0.374 ± 0.04 (P:0.01,T:2.51) | **61.014 ± 0.47** (P:0.01,T:2.60) | 0.364 ± 0.04 (P:0.01,T:2.69) | **63.141 ± 0.47** (P:0.00,T:2.85) | 0.557 ± 0.04 (P:0.00,T:11.74) | 40.538 ± 0.58 (P:0.00,T:-3.30) | 0.556 ± 0.04 (P:0.00,T:11.27) | 40.800 ± 0.58 (P:0.00,T:-3.20) |
> | Ours     | **0.377 ± 0.04**    | 61.285 ± 0.47        | **0.368 ± 0.04**    | 63.424 ± 0.47       | **0.562 ± 0.58**    | **40.474 ± 0.34**    | **0.561 ± 0.04**    | **40.735 ± 0.58**   |
>
>
> Regarding the experimental setup, please note that the XAI experiments are deterministic, so the seed has no impact. This methodology was also not employed in previous work [1–3], including papers accepted to ICML and CVPR. Regarding the number of examples, we thank the reviewer for this comment. The information was mistakenly removed from the paper before submission. We will add the number of examples used to each table. For better clarity, we report the number of examples used in each experiment here: For ARC-Easy, we use 5,200 samples; for Wiki-text, 5,000 examples; and for IMDB, 5,000 examples.
>
>
> >Q1 In Figure 3, the explainability heatmaps generated by PA-LRP and AttnLRP appear visually similar. Could you elaborate on the qualitative differences that PA-LRP captures and explain why these differences might not be visually apparent despite the claimed improvements?
>
>
> We want to clarify that the impact of PE on attribution is less significant for input-level PE (as used in ViT) than for attention-level PE (primarily employed in LLMs). Therefore, the differences are expected to be relatively minor in ViT. That said, we believe Figure 3 (see analysis in lines 302–323) clearly describes the differences between the explainability heatmaps. For example, see line 310:
>
> >> The attributed signal derived solely from positional-associated relevance captures unique relationships, exhibiting clearer spatial and structural patterns. In particular, relevance is distributed across the entire object, especially in the snake, bird, and shark examples. In contrast, the baseline method, which does not propagate relevance through PEs, produces a sparser pattern that does not focus on the entire object but instead is highly selective to specific patches. One possible explanation is that positional-associated relevance better captures concepts related to position, structure, order, and 321 broader regions within the image.

---

> > ### Comment · Reviewer_jDLx · 2025-08-06
> > **Thanks**
> >
> > I've read the authors' rebuttal, and they have fully addressed my concern regarding Q2 and partially addressed my concern regarding Q1. I will keep my score.

---

### Official Review · Reviewer_jV3L · 2025-06-30

**Clarity:** 2
**Significance:** 3
**Originality:** 2
**Rating:** 4
**Confidence:** 4

**Summary:**

This paper approaches the LRP based explainable methods for transformers and extend this method by assigning relevance for the positional encodings, one of the important and key components of the transformers. This paper proposes theoretically grounded relevance rules to propagate the attributions for the positional embeddings.

**Questions:**

1. One of the questions that arise are the situations where positional embeddings are less informative in the predictive performance for tasks such as classification?

**Ethical Concerns:**

["NO or VERY MINOR ethics concerns only"]

**Final Justification:**

Most of my concerns were addressed.

**Limitations:**

Yes, the authors did mention the limitations in Section 6.

**Paper Formatting Concerns:**

No, paper is correctly formatted.

**Quality:**

2

**Strengths And Weaknesses:**

Strengths:
1. One of the major strength of this paper is reformulation to include the positional embeddings theoretically.
2. The authors mention that assigning relevance to the positional parts can be decoupled and applied to any LRP based framework.
3. The empirical results shows the effectiveness of assigning the relevance to PE parts specially for the segmentation tests.

Weaknesses
1. This work is based on the premise that LRP based explainable methods are most effective model specific XAI method. It would be helpful if authors put references for this claim.
2. This work seems an incremental work on the AttnLRP with additional considerations of positional encodings. This simply appends the positional relevance to that of Attn LRP. The interactions between the attention layers and tokens needs to be explained.
3. In quantitative analysis, positive perturbation tests for DeIT variants shows Attn LRP performing better for Base and Tiny models. How does the removal of relevant pixels changes the positional understanding, eg: it is understandable that removing relevant pixels does changes the prediction performance but how does the relevance is distributed in context of the positional encodings and propagation rules.
4. The evaluation is mostly dependent on the perturbation and segmentation based tests. Human based tests might have helped. The quantitative test are mostly acts as proxies.

---

> ### Author Rebuttal · Authors · 2025-07-31
>
> >W.1. The premise that LRP based explainable methods are most effective model specific XAI method.
>
> While we acknowledge that no single XAI method can be definitively labeled as universally superior, we maintain that LRP shows the most promise for our specific application. Although this conclusion arises from the results of the Attention LRP paper, to support this position, we conducted comparative experiments against established methods including IG, SHAP, and [Leeman at al], which consistently demonstrated LRP's superior performance in our evaluation metrics. Please see Tables A and B mentioned in the response for reviewer jDLx.
>
> >W.2.1. Simply appends the positional relevance to that of Attn LRP.
>
> We respectfully disagree that our work is incremental. In the LRP literature, each paper focuses on improving different aspects. For example, [1], presented at ICML, focuses on softmax and LayerNorm, while attnLRP [2], also presented at ICML, focuses on attention layers. Our method revisits the concept of attributing PEs and propagating relevance through PE layers, and demonstrates both empirical and theoretical importance, especially for LLMs such as LLaMA3. We consider this a major advancement, as XAI tools for such models are extremely important for the community.
>
> > W.2.2. The interactions between the attention layers and tokens needs to be explained.
>
> Regarding the comment “The interactions between the attention layers and tokens need to be explained,” could you please elaborate? In Section 2.1, we discuss how attention is computed using RoPE and ALiBi, and in Section 3.2, we explain how relevance should be propagated through the RoPE and attention layers.
>
>
>
>
> >W.3. In quantitative analysis, positive perturbation tests for DeIT variants shows Attn LRP performing better for Base and Tiny models. How does the removal of relevant pixels changes the positional understanding, eg: it is understandable that removing relevant pixels does changes the prediction performance but how does the relevance is distributed in context of the positional encodings and propagation rules.
>
>
> The reviewer seems to ask what would be the effect of removing relevant pixels for prediction on the distribution of positional and semantic attribution maps. The question should be asked backwards, how do the positional and semantic attribution maps differ in terms of the order of the most relevant pixels.
>
> The conclusion should be drawn backwards, how does positional and semantic understanding (e.g. attribution maps) affect the resulting order of which pixels are most relevant for classification.
>
> We refer the reviewer to our qualitative results section (line 296), where we discuss the difference between positional and semantic attributions. In general, for a few cases, our approach demonstrates a minor decrease in positive perturbation performance while achieving substantial improvement in negative perturbation performance. These results mainly show that positional attribution facilitates identifying truly important features, but at the cost of giving higher relevancy scores to additional tokens, which may have been critical for the computational flow (for example for storing global information regarding the prediction), but are not by themselves crucial for prediction.
>
>
> >W4. The evaluation is mostly dependent on the perturbation and segmentation based tests. Human based tests might have helped. The quantitative test are mostly acts as proxies.
>
> We respectfully disagree with the reviewer, as there is no definitive and reliable way to evaluate XAI performance using human-based testing.
>
>
> >Q1. One of the questions that arise are the situations where positional embeddings are less informative in the predictive performance for tasks such as classification
>
> Our results strongly suggest that for most situations, positional embeddings are crucial for high-quality XAI performance. PE information is already thoroughly integrated throughout the model's representations and mixed with semantic representations. Attempting to isolate specific conditions where PE attribution might or might not be beneficial goes beyond the paper's scope.
>
> ___
>
> **References:**
>
> [1] XAI for Transformers: Better Explanations through Conservative Propagation. Ali et al. ICML 2022
>
> [2] AttnLRP: Attention-Aware Layer-Wise Relevance Propagation for Transformers. Achtibat et al. ICML 24
>
> [3] Transformer Interpretability Beyond Attention Visualization. Chefer et al. CPVR 2021

---

### Official Review · Reviewer_Qg1b · 2025-07-01

**Clarity:** 3
**Significance:** 1
**Originality:** 3
**Rating:** 3
**Confidence:** 3

**Summary:**

This paper presents and addresses a limitation in the explainability of transformer-based deep learning models: The handling of positional encodings (PEs) in Layer-wise Relevance Propagation (LRP) methods. The authors argue that existing LRP-based XAI ignores the role of positional encodings, violating the conservation property and obscuring important positional and structural attributions. To tackle the problem, they reformulate the input space and propose new propagation rules. The effectiveness of the approach is evaluated on vision and NLP tasks.

**Questions:**

Questions: Can you elaborate on the baselines used for comparison: What was used as the input for plain LRP? Did you backpropagate to E_i or E_i + P_k for the standard learned or sinusoidal embeddings.?

Why can't we just use LRP from the sum E_I + P_K (for standard additive embeddings)?

What was the motivation between just adding (as far as I understand) semantic and positional importance?

**Ethical Concerns:**

["NO or VERY MINOR ethics concerns only"]

**Final Justification:**

I am maintaining my prior assessment.

**Limitations:**

Yes

**Paper Formatting Concerns:**

Formatting: Figure 3 violates the margin.
Writing: Section Quantitative Analysis is a stub with only one sentence (l. 324)

**Quality:**

2

**Strengths And Weaknesses:**

## Strengths
* **New problem suggestion.** The paper lays out a previously unconsidered issue with LRP methods
* **Theoretical Foundation.** The authors provide formal analysis of their rules in Lemma 1 and 2. The proofs are referenced to the appendix but the statements in the main paper are clear.
* **Experiments span two domains and feature relevant models** such as DeiT variants and the Llama models in realistic domains with quantitative and qualitative metrics
* **Open-Source Implementation.** The paper provides a complete codebase, supporting replicability and adoption.

## Weaknesses

* **Limited Scope: Heavy reliance on Transformers and LRP**: The method is tightly coupled to LRP for Transformers. While the extension to other LRP variants is mentioned, its generality outside the AttnLRP framework is assumed. Broader applicability or adaptations to alternative XAI paradigms (e.g., SHAP, counterfactuals, or other propagation-based methods) are not explored.
To motivate this, the authors argue that LRP is the most reliable technique for transformers (l.31). I don’t think that the reality is that simple. Some current works, e.g., Leemann et al (2024) investigate surrogate model explanations for transformers such as SHAP and LIME and find their performance often comparable to LRP in common metrics such as insertion or deletion curves. Thus, I think the scope and potentially impact could be extended by generalizing the approach to other XAI techniques as well.

* **Motivation for specific rules** LRP decompositions are not unique for a given function. While the rules are presented, there is no explanation for why exactly the given form was chosen. Even for the simple additions, their can be 0-LRP, epsilon-LRP or even Gamma-LRP Rules (Montavon, 2019). There should be some discussion about alternative rules and why they were discarded.

* **Relevance and Signficance.** I don't fully understand the signficance of the distinction between positional and semantic importance yet. Why don’t we just redefine the input to be $E_k$ + $P_i$ where $E_k$ is the semantic token embedding and $P_i$ is positional embedding $i$, and backpropagate to this input quantity using LRP? This should fulfill the conservation properties and attribute to token k at position I, which is exactly what we want? The paper claims to manage to separate positional relevance from semantic relevance, but doesn't discuss why it matters in the end.

* **Some confusion about notation.** Writing of Equation (4) is a bit confusing. While the input space is a set of vectors in (3), here in (4) it seems to be a set of tuples, however the size of the tuple and which combinations are possible is still unclear to me. Please elaborate.

* **No measures of dispersion.** Unfortunately, the results in Table 1-3 feature no measures of dispersion making it hard to assess their significance.



------------------
**References**

Grégoire Montavon, Alexander Binder, Sebastian Lapuschkin, Wojciech Samek & Klaus-Robert Müller. Layer-Wise Relevance Propagation: An Overview, 2019.

Leemann, Tobias, Alina Fastowski, Felix Pfeiffer, and Gjergji Kasneci. Attention Mechanisms Don’t Learn Additive Models: Rethinking Feature Importance for Transformers." Transactions on Machine Learning Research. 2024.

---

> ### Author Rebuttal · Authors · 2025-07-31
>
> We thank the reviewer for their detailed comments and suggestions and provide responses to the raised concerns below.
>
>
> > W.1. Limited Scope.
>
> We respectfully disagree with this statement. Transformer explainability, especially for LLMs, is a crucial emerging discipline, with LRP-based methods being at the center of this domain. Moreover, Attn-LRP [1] shows that LRP-based XAI methods consistently outperform other XAI methods such as KernelSHAP, IG, and others. We underline that at least two dozen papers proposing LRP techniques for transformers have been previously published at top-tier conferences, such as ICML and ICLR. Notable examples include [1-3].
>
> To establish LRP's superiority in XAI performance, we conduct a series of experiments using the baselines suggested by the reviewer (Leemann et al, IG, SHAP). The results, presented in Tables A and B, mentioned in the response to reviewer jDLx,, show that our method consistently outperforms the baselines. It is demonstrated that methods which are not LRP-based consistently show inferior performance across tasks. Moreover, our method consistently outperforms Attn-LRP, with no single case in which Attn-LRP achieves the best score. In particular, our approach improves the AU-MSE score in the generation scenario by 14.5% for LLaMA 2–7B, 10.6% for LLaMA 2–7B Quantized, and 51.41% for Tiny-LLaMA.
>
>
>
>
> Additional rows for Table 3 from our paper (perturbations in NLP under the finetuning regime), including variance values (shown after the ± symbol). The table reports paired t-test results comparing each baseline to our method, such that each element in the pair  the comparison is made for the classification accuracy, for each sample, across all perturbation steps. Results are based on 5,000 randomly sampled reviews from the IMDB test set. Please see Table A in response to reviewer jDLx.
>
> Additional rows for Table 5 from our paper (perturbations in NLP under the zero-shot regime), including variance values (shown after the ± symbol). For multiple-choice question setting, we utilize the ARC-easy dataset, containing 5200 samples. For next token prediction task setting, we use the Wiki dataset, containing 5000 uniformly selected samples . Please see Table B in the response for reviewer jDLx.
>
> We also note that SHAP is also significantly more computationally involved due to their model-agnostic nature, which requires surrogate modeling and repeated inference, resulting in a total runtime increase of approximately 5 times more  in our experiments.
>
> > W.2. Motivation for specific rules LRP.
>
>
> While exploring optimal rules for PE-LRP through theoretical analysis or grid search could further improve our empirical results, we deliberately chose not to perform hyperparameter tuning for our PE-LRP rules in order to neutralize the impact of hyperparameter search.
>
> In our experiments, we followed the same guidelines reported as optimal in AttnLRP, specifically a combination of the $\epsilon$-rule and the $\gamma$-rule. We agree with the reviewer that a broader discussion of alternative relevance propagation rules is warranted. To address this concern, we now include quantitative results on perturbation tests in vision tasks using the alpha-beta rule, demonstrating the significant performance degradation it causes in our setup, which is based on Attn-LRP.
>
>
> **Table C:** The table demonstrates that using the settings presented in AttnLRP (epsilon, gamma) provides the optimal performance for both methods. We underline that while the alpha-beta rule manages to achieve higher performance for negative perturbation for our method, it is accompanied by a drastic tradeoff with positive perturbations, which implies that the attribution map is coupled with high level of noise.
>
> | M. Size | Method                         | Negative ↑ (Predicted) | Negative ↑ (Target) | Positive ↓ (Predicted) | Positive ↓ (Target) |
> |---------|--------------------------------|----------------------|-------------------|----------------------|-------------------|
> | Base    | AttnLRP (α, β)                 | 44.612               | 40.542            | **9.172**            | 8.515             |
> | Base    | AttnLRP (ε, γ)                 | **52.185**           | **47.516**        | 10.784               | **8.032**         |
> |         |                                |                      |                   |                      |                   |
> | Base    | Ours (α, β)                    | **61.974**           | **55.237**        | 43.557               | 39.499            |
> | Base    | Ours (ε, γ)                    | 54.970               | 50.174            | **9.918**            | **9.237**         |
> |||||||
> | Small   | AttnLRP (α, β)                 | 45.698               | 40.890            | **9.822**            | **9.048**         |
> | Small   | AttnLRP (ε, γ)                 | **50.662**           | **45.105**        | 10.511               | 9.761             |
> |         |                                |                      |                   |                      |                   |
> | Small   | Ours (α, β)                    | **62.963**           | **55.114**        | 33.651               | 30.158            |
> | Small   | Ours (ε, γ)                    | 53.482               | 47.948            | **9.135**            | **8.477**         |
> |||||||
>
>
>
>
>
>
> > W.3. … distinction between positional and semantic importance  …
>
>
> We do not claim to fully disentangle positional relevance from semantic relevance. Rather, our aim is to highlight the existence of a positional relevance component—one that has been largely overlooked but is shown through our analysis to be crucial for achieving strong XAI performance. We first want to clarify that in our experiments, we included a separate setting ("PE Only") as an ablation to better understand and isolate the contribution of position-associated relevance, and in practice, both positional and semantic relevance were computed through the same LRP backward path.
>
> Secondly, we believe that the reviewer oversimplifies the problem. For example, in attention-level PE, such as RoPE (popular in LLMs like LLaMA and Qwen), PE is integrated into all attention layers, and thus cannot be treated as part of the input. Additionally, this type of PE is relative rather than absolute, so simply redefining the input is not feasible. Moreover, without dedicated LRP rules, relevance conservation will be violated, as PE components involve many nonlinearities (for example, RoPE includes trigonometric functions). Our work addresses all these challenges with both theoretical and empirical analysis.
>
>
> > W.4. Some confusion about notation.
>
>
> Equation 3 presents the input space as defined in previous work, while Equation 4 introduces our reformulation. The intuition is simple: previous methods define the input space using only the word embeddings (a list of vectors in $\mathbb{R}^D$). We argue that this is not precise for Transformers, as PE is also part of the input. In short, a transformer with PE operates over ordered sequences rather than unordered sets, so defining the input space without position does not accurately reflect the problem. Therefore, the input space should be defined as token-position pairs, consisting of both the semantic information representing the words (as in prior work) and the positional information, which we explicitly include in our reformulation ($\forall k \in [K], j \in [L]: P_{j,k}$).
>
> > W.5. No measures of dispersion.
>
>
> We thank the reviewer for the suggestion, which we address with dedicated experiments. To ensure the robustness of our results, we report variance across runs and conduct paired t-tests to support statistical significance. The results will appear in the camera-ready version. See examples below for perturbation tests in NLP. We demonstrate that the variance is consistently low, and paired t-tests yield small p-values with t-scores aligned with performance trends.
>
> Please see Tables A and B mentioned in the response for reviewer jDLx
>
>
> > Q.1: … What … input for plain LRP? Did you backpropagate to E_i or E_i + P_k ...?
>
> We first want to clarify that our main contribution focuses on attention-level PE, not input-level PE. Attention-level PE is widely used in modern LLMs and has a strong impact on the behavior of each attention layer. In this setting, our results are the strongest, and our theoretical analysis stands out (see Lemma 3.3).
>
> Following this perspective, our contribution is to define LRP rules that enable relevance to be propagated through PE layers such as RoPE, ALiBi, and input-level PE.
>
> For input-level PE, which is the simpler case and not our main contribution, we define the input space as the concatenation of $E_i$​ and $P_k$​. We then propagate relevance to both components and sum the positive values along the corresponding feature dimensions.
>
>
> > Q.2: Why can't we just use LRP from the sum E_I + P_K (for standard additive embeddings)?
>
> First, the main part of our work, which is more complex, novel, and effective: our PE-LRP rules for attention-level PE used in LLMs.
> For input-level PE, simply applying LRP to the sum $E_i + P_k$​ can lead to a loss of relevance, as negative positional relevance may override positive semantic relevance at each coordinate, and vice versa. More importantly, since LRP is designed to propagate relevance all the way to the input space, stopping relevance propagation at the path embeddings in ViT models results in ignoring the pixel-space and producing explanations based only on the patch-space (and ignoring the entire computation of the patch embedding). This contradicts the core principle of LRP, which requires relevance to be propagated to the actual input.

---

### Official Review · Reviewer_xvHC · 2025-07-03

**Clarity:** 3
**Significance:** 3
**Originality:** 3
**Rating:** 4
**Confidence:** 3

**Summary:**

The work proposes an extension to Layer-wise relevance propagation (a feature
attribution approach) for Transformer models, which encompasses rules to
explicitly assign importance to positional embeddings. Specifically, the work proposes rules for five approaches to positional embedding: learnable, sinusoidal, rotary, and relative positional embeddings, as well as ALiBi.
The work provides some theoretic motivation regarding the conservativity and fidelity of the approach compared to AttnLRP.
The work compares the proposed approach to AttnLRP in empirical perturbation,
ablation, and segmentation experiments on vision data using DeiT, as well as on
natural language data using LLaMa-2-7B, LLaMa-2-7B quantized, Tiny LLaMa, and
LLaMa-3-8B.
The work also provides some qualitative comparison on both vision and language data.

**Questions:**

1. The quantitative experiments do not include any errors over the trials. I
   could also not find the number of trials conducted for these experiments.
   Regarding my following two questions, there might seem to be some cases in
   which there is no clear benefit from explicitly attributing positional
   embeddings. These points make the results somewhat inconclusive, as the work
   does not provide any analysis as to when the proposed approach should be
   used, and when it should not. The qualitative results also do not seem to
   give a clear difference. The work would be much more convincing if it could
   provide the number of trials, and conduct a Wilcoxon-Rank-Sum test to
   provide more conclusive evidence in favor of attributing positional
   embeddings. It would be immensely helpful for the community to understand
   when and why we should consider positional embeddings for LRP, to which I
   was not entirely able to convince myself based on the results.

2. For Table 1, the work states that the proposed approach lags behind for the
   positive perturbation, but performs better in the negative perturbation. This
   seems to suggest that for some setups, it is better to not regard the
   positional embedding. What are the implications of this observation? How
   significant are these results?

3. In Table 3, PE seems to outperform both AttnLRP as well as AttnLRP+PE on
   Tiny LLaMa. However, adding AttnLRP+PE does not seem to improve
   significantly over AttnLRP. Do you have any idea why this could be the case?

4. I am sceptical of how useful Lemma 3.1 is. As far as I understand, AttnLRP
   assumes the feature attribution only for the sum $R_E + R_P$, where the
   positional part is simply by design part of the input. I do understand that
   there are some limitation to not model the positional embedding explicitly,
   which this work empirically evaluates. However, I do not find this
   conservativity argument meaningful for understanding the distinction between
   handling and not handling positional embeddings. I also do not think that
   Lemma 3.2 is very useful, as Eq. 9 is per definition conservative.

5. Lemma 3.3 claims that in general, current LRP attribution rules (i.e.,
   AttnLRP) provide low faithfulness for attention-level PE transformers with
   respect to positional features. This lemma is not proven, but rather only a
   specific extreme-case example is provided. While the lemma could be
   reformulated, I do not think it makes a particularly convincing argument as
   to when it is useful to regard positional embeddings, and when it is not.


I would like to emphasize that I think it is helpful to consider positional
embeddings in LRP. However, I think that there are use-cases for which
positional embeddings do and do not matter. This works in its current state
seems to be convinced that positional embeddings should always be regarded,
although empirical experiments do not seem fully conclusive on this idea. The
work would be much stronger if it analyzed the idea that the usefulness of
positional information might be conditional, and explored cases for which it
might not. With the current state, I am leaning towards reject, but could be
convinced the empirical experiments would be analyzed for statistical
significance, as well as regard cases in which attributing positional
embeddings might not be useful.

Minor:
- I think it might be more descriptive to call the "Lemmas" propositions
  instead. Lemmas usually aid in the proof of some Theorem, which is not the
  case in this work.
- I think the notation in Appendix E is overly simplified, especially
  considering the sums without subscripts. I would also recommend to use the `\text{input}` etc.\ for descriptive subscripts.

**Ethical Concerns:**

["NO or VERY MINOR ethics concerns only"]

**Final Justification:**

With the authors' clarifications and updated error bars for the empirical experiments, as well as the statistical tests, assuming the authors intend to update all the results where this applies and supply the number of examples in the paper, I am confident to raise my score to Borderline Accept.

However, I currently do not feel confident to raise my score beyond this, as the theoretical analysis is lacking in formality and meaning.

**Limitations:**

The work states that it is not a limitation that it does not analyze existing LRP rules.
While I agree with this statement, I do not think it provides any insights into
the actual limitations of this work.
In its conclusion, the work describes a drastic improvement of XAI when
considering positional information. I think a more interesting limitation would be to describe cases where this positional information could be even misleading.

**Paper Formatting Concerns:**

No concerns.

**Quality:**

2

**Strengths And Weaknesses:**

### Strengths

- AttnLRP provides a powerful feature attribution approach. Improving this
  approach with an explicit handling of the positional embedding is a useful
  contribution to the field.
- The work touches a multitude of positional embedding approaches, which
  significantly aids in its practicality.
- The work uses established evaluation metrics for feature attribution, namely
  perturbation and ground-truth segmentation.
- The work follows a clear structure and is well-written.

### Weaknesses

- The empirical experiments are partly inconclusive, given that for some
  metrics, the baseline wins, and for others, the improvements seem to be
  marginal. However, there are no error bars or number of trials reported,
  which would aid in testing for statistical significance.
- The work argues that AttnLRP is not conservative unless positional embeddings
  are attributed. I do not understand why this argument matters for the
  distinction between considering and not considering positional embeddings.
  Even if one would deem this argument valuable, both approaches seem to be
  conservative in their own perspective.
- The work claims to present some theoretical evidence that the omission of
  positional embeddings in attribution gives low fidelity. However, this is not
  proven, but only an extreme-case example is provided. The claim is also not
  consistent with the empirical results, which indicate that in some cases,
  fidelity can be higher when omitting the attribution of positional
  embeddings.

---

> ### Author Rebuttal · Authors · 2025-07-31
>
> We are grateful for the reviewer’s feedback.
>
> > W.1 The empirical experiments are partly inconclusive, given that for some metrics, the baseline wins, and for others, the improvements seem to be marginal.
>
>
> We respectfully disagree with the reviewer’s assessment and maintain that our empirical results clearly demonstrate the importance of attributing positional components for achieving high XAI performance. In the NLP domain, using attention-level PE, our method consistently outperforms Attn-LRP across all model capacities and evaluation settings. Notably, on the IMDB classification task (Table 3), our approach improves the AU-MSE score in the generation scenario by a substantial margin, ranging from 14.5% to 51.41%.
>
> While we occasionally observe only marginal improvements on certain metrics, these cases are consistently accompanied by significant gains on complementary metrics for the same model capacity. In some perturbation experiments for vision tasks, our method exhibits a slight decrease in positive perturbation performance but consistently achieves substantial improvements in negative perturbation performance—highlighting its overall advantage.
>
>
>
>
>
> > W.1. & Q.1.1 there are no error bars or number of trials reported.
>
> Thank you for pointing out this shortcoming. We will add the number of runs, SD, and paired t-tests to all experiments. Reassuringly, all differences we pointed out in the paper are significant, verified using the paired t-tests. We report p<0.03 figures for our experiments, mostly with p values lower than 10^-4. See, for example, Tables A,B  below, which provide the analysis for Tables 3,5 in the paper, respectively.
>
>
>
>
>
>
>
> >W.2. … argues that AttnLRP is not conservative unless positional embeddings are attributed. I do not understand why …
>
> The conservation property is fundamental to LRP's interpretability guarantees, ensuring attribution magnitudes remain proportional to the function’s output, and is highly regarded as a crucial factor for achieving faithful explanations [1, 2, 3]. The conservation property by itself is not a consideration for the distinction between considering and not considering positional embeddings, but rather a theoretical justification to explain improvement for empirical results.
>
>
>
>
> > W.3. The work claims to present some theoretical evidence that the omission of positional embeddings in attribution gives low fidelity. However, this is not proven, but only an extreme-case example is provided. The claim is also not consistent with the empirical results, which indicate that in some cases, fidelity can be higher when omitting the attribution of positional embeddings.
>
>
> We respectfully disagree with the claim that our fidelity argument is not proven. It is supported by a counterexample, a well-established and effective proof technique, and follows the same standard used in prior work in the domain (e.g., Chefer, Lemma 3; AttnLRP, Appendix A.2.1). We also do not consider this proof to fall under the category of an “extreme case.” Rather, it reflects the simplest and most controlled setting, demonstrating that even in such basic scenarios, fidelity issues can arise.
>
>
> We would like to emphasize that employing simplifications to derive theoretical insights is a widely accepted practice in deep learning research. In particular, theoretical analyses of faithfulness in LRP methods (especially in the context of Transformers) are exceptionally rare and inherently difficult to formalize. From this perspective, we view Lemma 3.3 as a meaningful and technically valuable contribution.
>
>
> With regard to consistency with empirical results, we emphasize that our theoretical claims on faithfulness are solely focused on attention-level PE, which is widely used in LLMs. In the experiments we conducted with attention-level PE (see Tables 3 and 5), the results are highly consistent with our theory. For example, when incorporating our PE-LRP rules into LLaMA models, our approach improves the AU-MSE score in the generation scenario by 14.5% for LLaMA 2-7B, 10.6% for LLaMA 2-7B Quantized, and 51.41% for Tiny-LLaMA. Notably, AttnLRP never outperforms both of our methods (PE-LRP and PE only) in any setting (see Tables 3 and 5) in the attention-level regime.
>
>
> > Q.1.2. .. there might seem to be some cases in which there is no clear benefit from explicitly attributing positional embeddings. ..These points make the results somewhat inconclusive
>
> We clarify that our main results are not inconclusive. Our PE-LRP rules are particularly effective when incorporated in the attention-level, and in those settings (see Tables 3 and 5 in our paper), our method consistently outperforms Attn-LRP, with no single case in which Attn-LRP achieves the best score.
>
>
> > Q.1.3. These points make the results somewhat inconclusive, as the work does not provide any analysis as to when the proposed approach should be used, and when it should not. The qualitative results also do not seem to give a clear difference…
>
>
> With regard to determining when and why to propagate PE is necessary, our empirical results demonstrate that PE attribution is crucial for explanation quality across the board - the 'when' is always for models with attention-level PE (our NLP experiments) and almost always for Vision, and the 'why' is that PE information is already thoroughly integrated throughout the model's representations and therefore must be attributed. Attempting to isolate specific conditions where PE attribution might or might not be beneficial goes beyond the paper's scope.
>
>
> > Q.2. For Table 1, the work states that the proposed approach lags behind for the positive perturbation, but performs better in the negative perturbation. This seems to suggest that for some setups, it is better to not regard the positional embedding. What are the implications of this observation? How significant are these results?
>
>
> The reviewer misinterprets the mixed positive/negative perturbation results reported for vision classification task (Table 1). For a few cases, our approach showed a minor decrease in positive perturbation performance while achieving substantial improvement in negative perturbation performance. This trade-off is favorable - the slight positive perturbation decrease is outweighed by the significant negative perturbation gains, representing an overall improvement in explanation fidelity. These cases do not support the reviewer's broader claim about settings where disregarding positional embeddings is beneficial; rather, they demonstrate our method's effectiveness. These results mainly show that our approach is better at identifying truly important features, but it may have some noise in its ranking.
>
> Additionally, it is important to note that our work does not perform hyperparameter tuning for the LRP hyperparameters or the selection of LRP rules, and thus our setup is likely not optimal and could be readily improved through such procedures.
>
>
> > Q.3. In Table 3, PE seems to outperform both AttnLRP as well as AttnLRP+PE on Tiny LLaMa. However, adding AttnLRP+PE does not seem to improve significantly over AttnLRP. Do you have any idea why this could be the case?
>
>
> One possible explanation is that in Table 3, when evaluating XAI performance for sentiment analysis, the model relies heavily on opening statements and closing conclusions to make its prediction. As such, most of the prediction is attributed to the positional components of tokens at the beginning and at the end of the prompt.
>
>
>
>
> > Q.4.1. I am sceptical of how useful Lemma 3.1 is...
>
>
> For Lemma 3.1, referring to input-level PE, such as in vision transformers, the image patches undergo convolution (or linear layers)  first, then PEs are added to these patch embeddings before entering the transformer layers. This means PE is not simply "part of the input by design" - it's a separate additive component that AttnLRP treats as transparent. Moreover, while some methods choose to focus only on attributing the sum of positional and semantic relevance, without propagating relevance through the entire model as LRP is designed to do, this decision is unjustified and purely heuristic. Our approach provides a clear rationale for this design choice and defines how positional encodings should be treated.
>
> > Q.4.2. However, I do not find this conservativity argument meaningful for understanding the distinction between handling and not handling positional embeddings.
>
> Assume, for example, that in a ViT model, patch embeddings (denoted by E) are extracted via a convolutional layer, and then PEs (P) are added to form Z=E+P, which is then passed into the Transformer. If we aim to propagate relevance back to the input space of the model at the pixel level (what LRP is designed for), the existing methods typically propagate relevance only through E, not through P. As a result, the relevance R(E) is propagated through the convolution back to the input, but any positional relevance R(P) is lost.
>
> Additionally, for input-level PE, simply applying LRP to the sum positional and semantical relevance​ can lead to a loss of relevance, as negative positional relevance may override positive semantic relevance at each coordinate, and vice versa.
>
> > Q.4.3. .. do not think that Lemma 3.2 is very useful, as Eq. 9 is per definition conservative.
>
> While there are specific cases (due to a combination of architecture, particular LRP rules, and the unjustified decision to stop relevance propagation before reaching the input space) where ignoring PE does not violate conservation, there are also clear cases where conservation is violated, such as the example discussed in our response to Q.4.2. In general, when relevance is propagated all the way to the input space rather than stopping at the point where PE is added, ignoring PE leads to a violation of conservation. This is critical, as propagating relevance to the final input space is a fundamental principle of LRP.

---

> > ### Comment · Reviewer_xvHC · 2025-08-05
> >
> > Thank you for your clarifications, and the updated tables.
> >
> > ## Q1, W1
> >
> > With the error bars,
> > which were also requested by the other reviewers,
> > I am convinced by the empirical results.
> >
> > ## Q2
> >
> > Thank you for the clarification.
> > I can see that there is an overall improvement.
> >
> > ## Q3
> >
> > Thank you for your intution.
> > If the feature attribution requires a primary focus on the specific location,
> > then I would also expect that for vision setups,
> > where the object of interest is mainly in the center of the image,
> > the positional embedding part alone could already give good performance.
> > Is there a specific reason why these results
> > were not included in Tables 1 and 2?
> >
> > ## Q4.1, W2
> >
> > Treating the PE part as transparent
> > and simply attributing the full relevance to the input
> > will also results in conservativity.
> > The important point that the work is trying to make,
> > and I do not think Lemma 3.1 captures this very well,
> > is that under the assumption
> > that the PE should not be treated as transparent,
> > AttnLRP is non-conservative.
> > I think this is precisely the reason why Reviewer Qg1b asks
> > why we cannot simply attribute the some of $R_E + R_P$.
> >
> > But once you make the assumption clear
> > that the PE should not be treated transparently,
> > then non-conservativity trivially follows,
> > which is why I do not think Lemma 3.1 is very useful.
> >
> > Or in other words:
> >
> > Assuming a decomposition of the input $R_\text{input}$
> > into token $R_E > 0$
> > and positional embedding $R_P > 0$ (because we do not treat it transparently),
> > $R_\text{input} = R_E + R_P$,
> > it follows by definition that $R_\text{input} > R_E$.
> >
> >
> > ## W3
> >
> > The claim in Lemma 3.3 is extremely general and unspecific. It states:
> >
> > > For attention-level PE transformers, current LRP attribution rules achieve low faithfulness, especially when considering positional features.
> >
> > I doubt this claim alone can be proven in any meaningful way at all,
> > as it is not stated when low-faithfulness is achieved
> > (always?, in specific cases?;
> > also, low-faithfulness is not well defined, but that is besides the point).
> > By reading the authors' clarifications,
> > I believe that the claim the authors are trying to proof by example is:
> >
> > *There exists a setting for which Attn-LRP produces low faithfulness when considering positional encodings.*
> >
> > For which the proof in the appendix would be sufficient.
> > This statement can be helpful to understand
> > that there exists a setting at all where
> > considering positional embeddings improves fidelity.
> > However, on its own,
> > it does not make any general statements on
> > whether it makes sense to consider PEs
> > (e.g., it could make things worse in some settings,
> > but we do not know without proving anything).
> >
> > A much more useful and general proof would be to show
> > that for some specific setup,
> > considering PEs leads to a fidelity
> > larger equal the fidelity when not considering PEs.
> >
> > The specific claim that Lemma 3.3 is serving to prove is found in line 244:
> >
> > > In particular, we show that within simplified settings, LRP yields unfaithful explanations when the task relies heavily on positional features, such as predicting the number of tokens.
> >
> > As far as over-claiming goes,
> > the only problematic part of this is that
> > "simplified settings" should be
> > "in a specific simplified setting".
> >
> > But thinking about the goal of the paper,
> > the current theoretical analysis
> > with the exemplary setting for Lemma 3.3
> > is not giving a lot of insight.
> >
> > > We would like to emphasize that employing simplifications to derive theoretical insights is a widely accepted practice in deep learning research.
> >
> > I do not disagree with the authors at all about the value
> > of simplified settings for theoretical insights in deep learning.
> > But I urge the authors to not abuse simplification
> > as an argument to justify a lack of formal and meaningful statements.
> >
> >
> > ## Conclusion
> >
> > With the authors' clarifications
> > and updated error bars
> > for the empirical experiments,
> > as well as the statistical tests,
> > assuming the authors intend to update
> > all the results where this applies
> > and supply the number of examples in the paper,
> > I am confident to raise my score to Borderline Accept.
> >
> > However,
> > I currently do not feel confident to raise my score beyond this,
> > as the theoretical analysis is lacking in formality
> > and meaning.

---

### Note · Authors · 2025-08-13

**Dear Area Chair and Reviewers,**

We sincerely thank all reviewers for their thoughtful engagement with our work. Following extensive discussions, we believe we have successfully addressed the primary concerns raised.

**Statistical Significance (All Reviewers):** We have added comprehensive statistical analysis including variance measures, paired t-tests, and sample sizes across all experiments. The results demonstrate strong statistical significance ($p<0.03$, often $p<10^{-4}$) for our improvements, particularly in attention-level PE settings where our method improves AU-MSE scores by 14.5-51.41% across different models.

**Theoretical Contributions (Reviewers xvHC, Qg1b):** Our theoretical analysis, using simplified settings as is standard in the field, provides meaningful insights. We prove that ignoring PE violates conservation properties and leads to unfaithful explanations. In our work, we have presented fundamental issues that arise even in basic scenarios. We underline that the conservation property is crucial for LRP's interpretability guarantees, as established in prior literature.

**Scope and Impact (Reviewer Qg1b):** We respectfully maintain that transformer explainability represents a crucial research direction, with LRP-based methods being at the center of this domain. Our comparative experiments against SHAP, IG, and other methods, confirm LRP's superior XAI performance. We report that our PE-LRP rules lead to the strongest performance, particularly for attention-level PE.

**Technical Novelty (Reviewer jV3L):**  Our contribution extends beyond simply "appending" PE to AttnLRP. Our contribution consists of non-trivial design choices including: (1) defining PE sinks and reformulating the input space, (2) selecting appropriate PE representations that preserve conservation, and (3) implementing a multi-sink strategy validated through ablations. We additionally develop tailored rules for multiple PE variants (RoPE, ALiBi, learned, sinusoidal), each requiring specific theoretical treatment.

**Empirical Results:** Our method consistently outperforms baselines in attention-level PE settings. While improvements in input-level PE (vision) are more modest, this aligns with our theoretical predictions about where PE attribution matters most.

We believe our work makes a significant contribution by identifying and addressing a critical gap in transformer explainability methods, with both theoretical grounding and strong empirical validation.

---

### Decision · Program_Chairs · 2025-09-17

**Decision:**

Accept (poster)

**Comment:**

This paper proposes an extension of AttnLRP, one of the best performing explanation methods for transformer models. The extension consists of introducing explicit positional embeddings to input and proposing dedicated rules to explicitly assign importance to these embeddings. This extension is theoretically well-motivated (e.g., increase the conservativity and fidelity of the approach) and results in good explanations. The method is extensively evaluated on different datasets and with ablation studies.
The strength of the paper is that it contributes a novel and important insight, namely the importance of taking positional embeddings into account when explaining predictions of transformer models. This insight is theoretically well-motivated and results in good explanations. Experiments are convincing, spanning two domains and sota models in realistic settings. On the negative side, reviewers criticize the narrow focus on LRP and the inconclusiveness of the experimental results (e.g., suggestion for human-based evaluation).
Some of the issues raised by the reviewers could be resolved in the rebuttal. Overall, the reviewers are on average slightly more on the positive side, stressing the importance and potential large impact of the key message of the paper (i.e., role of positional encodings). Since AttnLRP is a state-of-the-art approach for explaining transformers and this topic is of high relevance, even a small improvement (here it also comes with a theoretical motivation) is valuable. Therefore, I recommend to accept the paper.